# Study of the seasonal variation of Aeolus wind product performance over China using ERA5 and radiosonde data

Siying Chen[1], Rongzheng Cao[1], Yixuan Xie[1], Yinchao Zhang[1*], Wangshu Tan[1], He Chen[1], Pan Guo[1], and Peitao Zhao[2*]

[1]School of Optics and Photonics, Beijing Institute of Technology, Beijing 100081, China
[2]Meterological Observation Center of CMA, Beijing 100081, China

*Correspondence to*: Yinchao Zhang (ychang@bit.edu.cn) and Peitao Zhao (peitaozhao@163.com)

**Abstract.** Aeolus wind products became available to the public on May 12, 2020. In this study, Aeolus wind observations, L-band radiosonde data, and the European Centre for Medium-Range Weather Forecasts fifth generation atmospheric reanalysis data were used to analyze the seasonality of Aeolus wind product performance over China. Based on the Rayleigh-clear and Mie-cloudy data, the data quality of the Aeolus effective detection data was verified, and the results showed that the Aeolus data were in good agreement with the L-band RS and ERA5 data. The Aeolus data relative errors in the four regions (Chifeng, Baoshan, Shapingba, and Qingyuan) in China were calculated based on different months (July to December 2019 and May to October 2020). The relative error of the Rayleigh-clear data in summer was significantly higher than that in winter, with the mean relative error parameter in July 174% higher than that in December. The mean random error increased 0.97 m/s in July compared with December, which also supported this conclusion. In addition, the distribution of the wind direction and high-altitude clouds in different months (July and December) were analyzed. The results showed that the distribution of the angle between the horizontal wind direction of the atmosphere and the horizontal line of sight, had a greater proportion in the high error interval (70°–110°) in summer, and this proportion was 8.14% higher in July than in December. The cloud top height in summer was approximately 3–5 km higher than that in winter, which might decrease the signal-to-noise ratio of Aeolus. Therefore, the wind product performance of Aeolus was affected by seasonal factors, which might be caused by seasonal changes in wind direction and cloud distribution.

## 1 Introduction

Global wind field data are indispensable meteorological parameters for weather forecasting (Ishii et al., 2017). The European Space Agency (ESA) proposed the Atmospheric Dynamics Mission Aeolus (ADM-Aeolus) in 1999. The Aeolus is equipped with a 355 nm direct-detection wind lidar, which uses a single-view detection method to get the horizontal line of sight (HLOS) component of the three-dimensional wind field from space. In addition, it adopts a dual-channel design and uses different frequency discriminators to receive the Mie and Rayleigh channel signals (Stoffelen et al., 2005; Reitebuch, 2012). Aeolus was successfully launched on August 22, 2018, becoming the world's first spaceborne wind lidar in orbit. It then sent back the first batch of wind profile data, proving that the satellite-borne direct-detection wind lidar could provide global wind profiles (Reitebuch et al., 2019). On May 12, 2020, ESA opened Aeolus's wind measurement products to the public (https://earth.esa.int/eogateway/news/aeolus-data-now-publicly-available, last access: March 13, 2021), including Level 1B (L1B) and L2B products. Among these, L2B products provide fully processed HLOS wind profiles after correction of temperature and pressure effects (European Space Agency et al., 2008), which were used in the present study.

To accurately calibrate the Doppler lidar carried on Aeolus, dedicated calibration and validation team have carried out a series of verification and comparison studies on Aeolus' wind products. After the launch of Aeolus, a special verification study was immediately performed. The main verification methods have included radiosonde (Baars et al., 2020) and airborne lidar (Lux et al., 2020; Witschas et al., 2020), in which Aeolus's prototype atmospheric laser doppler instrument airborne demonstrator was used. Over the following two years, researchers worldwide have completed regional verification of the Aeolus detection

data using various detection methods, including satellites (Shin et al., 2020), ground-based lidar (Hauchecorne et al., 2020), ground-based wind profiler radar (Belova et al., 2021; Guo et al., 2021), and radiosonde (Liu et al., 2021). In addition, global Aeolus data verification studies using the numerical weather prediction (NWP) model have been undertaken (Rennie and Isaksen, 2019; Martin et al., 2021). These verifications have deepened our understanding of Aeolus data quality, with some factors discovered that affect Aeolus data quality during the verification process, such as solar background radiation (Zhang

et al., 2020), satellite flight direction (Guo et al., 2021), and seasonal changes (Martin et al., 2021). The European Centre for Medium-Range Weather Forecasts (ECMWF) revised the data processing algorithm over time, including the temperature gradient correction algorithm for M1, which is the main mirror of the Aeolus telescope, to solve the problem of seasonal changes (Rennie and Isaksen, 2019).

To date, very few studies on the seasonal fluctuation of Aeolus wind product performance over China have been undertaken,

especially using actual detection data. After the implementation of the new M1 deviation correction scheme, the effect of system thermal performance changes on Aeolus' seasonal fluctuation is significantly reduced, which led to systematic errors lower than 1 m/s (Rennie and Isaksen, 2020). However, the actual atmospheric conditions during different seasons still affect the detection performance of the lidar.

In the present study, the variation of the Aeolus wind product performance during different seasons was analyzed using Aeolus

L2B wind products in four regions of China over 12 months (July to December 2019, and May to October 2020), and was compared to the L-band radiosonde (L-band RS) detection and ECMWF Reanalysis v5 (ERA5) reanalysis data. Two conjectures regarding Aeolus's different wind product capabilities were introduced and verified.

## 2 Data and methods

### 2.1 Aeolus L2B wind products

A sun-synchronous dawn-dusk orbit with a height of approximately 320 km is selected by Aeolus, and the orbit repeats its ground track every seven days (European Space Agency et al., 2008). The transit time of Aeolus over central and eastern China is at approximately 10:00 UTC and 22:00 UTC. In the present study, the criteria for judging the validity of the Aeolus data were the validity flags (0 = invalid and 1 = valid) and the estimated errors (threshold requirements), which were obtained from the L2B data. Based on the recommendations of the ECMWF (Rennie and Isaksen, 2020) and the actual data situation used in

the present study (Fig. 1), the thresholds of the estimated errors were 8 m/s (Rayleigh-clear) and 4 m/s (Mie-cloudy). In this study, the Mie-clear and Rayleigh-cloudy data were discarded because the remaining valid data points were too few.

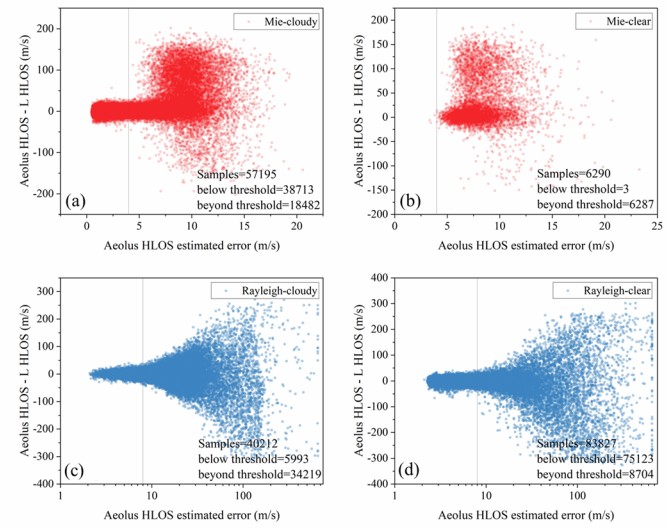

**Figure 1. Difference between the Aeolus HLOS and L HLOS wind components as a function of estimated errors for (a) Mie-cloudy, (b) Mie-clear, (c) Rayleigh-cloudy, and (d) Rayleigh-clear. Samples mean the number of data points and the samples below or beyond**
**the threshold are also listed. Reference lines are the screening threshold of estimated errors: 4 m/s (Mie) and 8 m/s (Rayleigh).**

## 2.2 L-band radiosonde wind data

The L-band RS is widely used to obtain the true situation of the atmospheric environment, and its detection altitude reaches 30 km (Guo et al., 2016). In the present study, the L-band RS wind data were the valid data detected by the four L-band RS stations. Matching the geographical location and time of the L-band lidar required special attention. For the geographical location, most of the valid L-band RS detection data used in the present study had a balloon drift of less than 0.5° (longitude or latitude). Only a few data points in winter had a maximum balloon drift of approximately 1.6°. The detection time of the L-band RS network in China was 0:00 UTC and 12:00 UTC. Generally, the time of launching the ball is approximately 45 min earlier than the detection time, which is 1 to 2 h away from the transit time of Aeolus. To reduce the influence of time and geographical location differences in the present study, ERA5 data were used as the reference data.

## 2.3 ERA5 data

The reanalysis dataset is often used as a reference for meteorological data analysis (Hersbach et al., 2020). ERA5 data provided by ECMWF were used in the present study as the reference data. The current observation dataset used for ERA5 assimilation does not contain the observation results of Aeolus ([https://confluence.ecmwf.int/display/CKB/ERA5%3A +data+documentation#ERA5:datadocumentation-TheIFSanddata assimilation/](https://confluence.ecmwf.int/display/CKB/ERA5%3A+data+documentation#ERA5:datadocumentation-TheIFSanddata assimilation/), last access: March 13, 2021); therefore, there was no mutual influence between the L2B data of Aeolus and the reanalysis data of ERA5. In addition to the zonal wind vector $u$ and the meridional wind vector $v$, the cloud coverage provided in ERA5 was used in the present study (Section 3.2.2). Because of the high resolution of ERA5 regarding time and geographical location, after matching well with Aeolus data, the difference between ERA5 and L-band RS data was used to represent the wind difference between Aeolus and L-band RS due to time and geographic differences to a certain extent.

## 2.4 Data matching

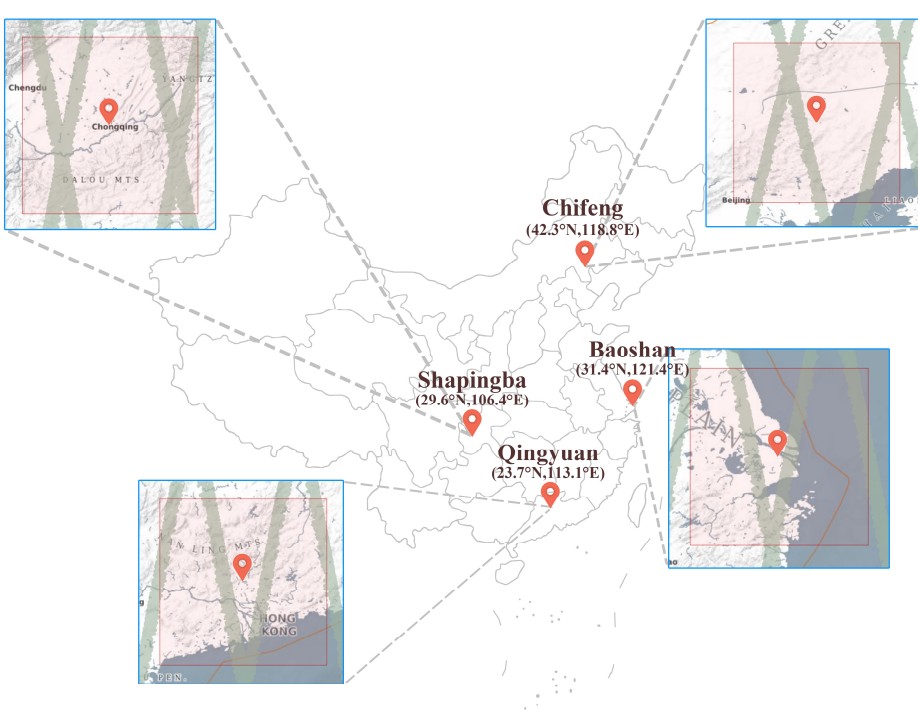

**Figure 2.  Geographical location of the L-band RS site and Aeolus measurement trajectory.  In the enlarged view ([http://aeolus-ds.eo.esa.int/socat/L1B_L2_Products](http://aeolus-ds.eo.esa.int/socat/L1B_L2_Products)), the red rectangles represent the range of ± 2.5° around the L-band RS stations, and the gray-green lines represent Aeolus's overlapping trajectories for the 12 months.**

95  The collocation distance suggested by the cal/val implemtation plan ([https://earth.esa.int/eogateway/documents/20142/1564626/Aeolus-Scientific-CAL-VAL-Implementation-Plan.pdf](https://earth.esa.int/eogateway/documents/20142/1564626/Aeolus-Scientific-CAL-VAL-Implementation-Plan.pdf) ) is 100 km around ground site, but for a single site, there are too few data points that meet the suggestion per month. To compromise the consistency of meteorological conditions and abundance of detection data, the Aeolus L2B data were compared with the ERA5 and L-band RS detection data, within a ± 2.5° (latitude and longitude) geographical range (Fig. 2) near the target L-band RS station. The four L-band RS stations shown

100  in Figure 1 were selected because they possessed obvious differences in geographical location and meteorological conditions. The main characteristics of the datasets used in the comparison are listed in Table 1. The main processing procedures are shown in Figure 3, and more details are discussed in follow.

| Data | Aeolus | ERA5 | L-band RS |
|---|---|---|---|
| Time | 2019.07–2019.12 and 2020.05–2020.10 Approximately 10:00 UTC and 22:00UTC | 2019.07–2019.12 and 2020.05–2020.10 Every hour | 2019.07–2019.12 and 2020.05–2020.10 Approximately 0:00 UTC and 12:00 UTC |
| Geographical location | ± 2.5° (latitude and longitude) near the target RS station | ± 2.5° (latitude and longitude) near the target RS station | Chifeng (42.3° N, 118.8° E) Baoshan (31.4° N, 121.4° E) Shapingba (29.6° N, 106.4° E) Qingyuan (23.7° N, 113.1° E) |
| Vertical resolution | 0.25 km to 2 km | 37 pressure levels for 1000 hPa to 1 hPa | < 10 m |
| Max. altitude | Approximately 20 km for valid data | Approximately 45 km | Approximately 32 km |

Table 1. Main characteristics of the datasets used in the comparison

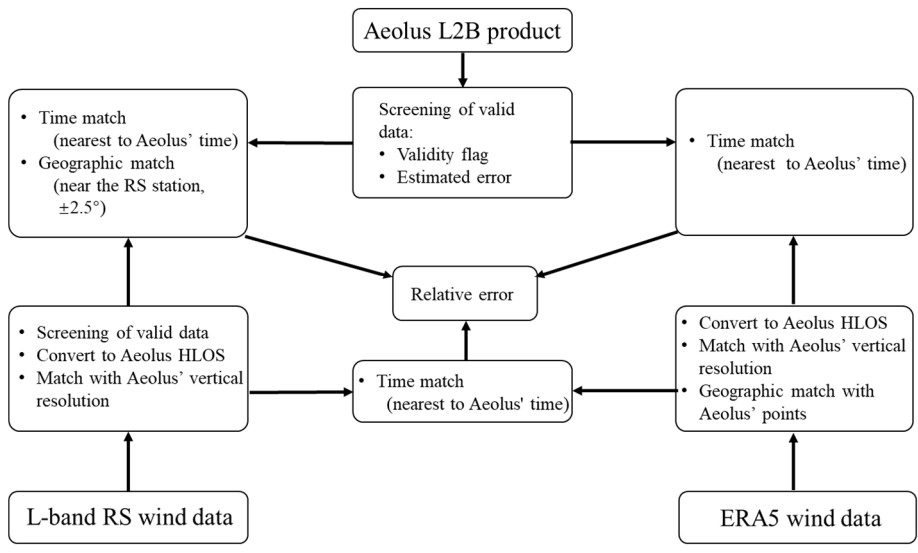

Figure 3. Main processing procedures of Aeolus, L-band RS, and ERA5 wind data

The L2B data is the result of the single-component wind measurements of Aeolus; therefore, it was necessary to decompose the L-band RS and ERA5 data in the direction of the Aeolus HLOS. The Aeolus has different azimuths at different locations and distances, which have been given in the L2B data. RS and ERA5 data were decomposed in the HLOS direction:

$$V_{HLOS} = V_{L/ERA5} \times \cos(\theta_{HLOS} - \theta_{L/ERA5}), \tag{1}$$

where, $V_{L/ERA5}$ represents the total horizontal wind speed provided by the L-band RS data or ERA5 data. $\theta_{L/ERA5}$ represents the wind direction of the total horizontal wind vector. $\theta_{HLOS}$ is the azimuth of Aeolus. The ERA5 wind data have full $\boldsymbol{u}$ and $\boldsymbol{v}$ wind victor information and can calculated the $V_{HLOS}$ through two components. However, the result is the same as that calculated by Eq. (1) after total wind vector composited.

For time matching, L-band RS and ERA5 take the latest detection data from the Aeolus transit to the target area ($\pm 2.5°$ near the L-band RS station). For geographical location matching, it was determined by selection of the L-band RS station. The Aeolus data selects the detection data within a $\pm 2.5°$ (latitude and longitude) rectangular area centered on the L-band RS station (red rectangular area in Fig. 2). For the ERA5 data, the data point closest to the latitude and longitude of the Aeolus data was selected. In the vertical direction, owing to the difference in the vertical resolution of the three datasets, L-band RS and ERA5 data had to be matched with the Aeolus data via linear interpolation. Linear interpolation is a method of curve fitting using linear polynomials to construct new data points within the range of a discrete set of known data points. The L-band RS (ERA5) data point, which just little higher than one Aeolus data point in altitude, was found and marked as $V(H^+)$. Then, $V(H^-)$ was found, which was just little lower than the Aeolus data point. The L-band RS (ERA5) wind matched with the Aeolus data point was calculated using Eq. (2):

$$V_H = V(H^-) + \frac{H_{Aeolus} - H^-}{H^+ - H^-} \cdot \left(V(H^+) - V(H^-)\right), \tag{2}$$

where, $V_H$ is the L-band RS (ERA5) wind matched with the Aeolus data at altitude, and $H_{Aeolus}$ is the altitude of the Aeolus data point.

**2.5 Calculation of the relative error**

Generally, the calculation formula of the wind speed error for wind measurement system is:

$$D = V_a - V_b, \tag{3}$$

where, $V_a$ and $V_b$ represent the wind speeds of different datasets. This equation was used to calculate the difference between the different wind field data. Then, for a dataset with a sample size of $n$, its statistical mean error (MD) and standard deviation (SD) were:

$$MD = \frac{1}{n}\sum_{i=1}^{n} D(i), \tag{4}$$

$$SD = \sqrt{\frac{1}{n-1}\sum_{i=1}^{n} (D(i) - MD)^2}, \tag{5}$$

In addition, the scaled median absolute deviation (scaled MAD) is widely used in other Aeolus validation studies:

$$\text{scaled MAD} = 1.4826 \times \text{median}(|D - \text{median}(D)|), \tag{6}$$

However, because the detection error of the Doppler wind lidar increases with an increase in the detected wind speed (Frehlich, 2001), the relative error can better reflect the detection performance of the instrument compared to the error value. The calculation of the relative error is generally shown in Eq. (7):

$$D_r = \frac{V - V_{ture}}{V_{ture}}, \tag{7}$$

where, $V_{ture}$ represents the data with a smaller error in the two datasets. In the present study, it was assumed that the error of ERA5 data was the smallest, while the error of Aeolus data was the largest.

Similarly, for a dataset with a sample size of $n$, the statistical average relative error was:

$$\bar{D}_r = \frac{1}{n}\sum_{i=1}^{n} |D_r(i)|, \tag{8}$$

In the data comparison logic of the present study, both $\bar{D}_r(Aeolus\&L) - \bar{D}_r(L\&ERA5)$ and $\bar{D}_r(Aeolus\&ERA5)$ were used to approximate the relative error value of the Aeolus data when the space-time matching was good. The Aeolus relative error parameter was set as follows:

$$D_{Aeolus} = \frac{[\bar{D}_r(Aeolus\&L) - \bar{D}_r(L\&ERA5)] + \bar{D}_r(Aeolus\&ERA5)}{2}, \tag{9}$$

150     The average value of the comparisons between Aeolus and the two datasets was used to reduce the possibility of large deviations in the relative error. $D_{Aeolus}$ was used to approximate the relative error value of the Aeolus data in the present study.

## 3 Results and discussion

### 3.1 Data quality

The comparison results of the three data sources after data matching are shown in Figure 4. The red and blue points in the
155 figures represent the data points of the Mie-cloudy and Rayleigh-clear groups, respectively. The red and blue lines represent the linear fitting lines of the data points of the two groups, respectively. The Mie-cloudy and Rayleigh-clear groups in the four regions provided 38,275 and 73,131 valid data points over 12 months (July–December 2019 and May–October 2020).

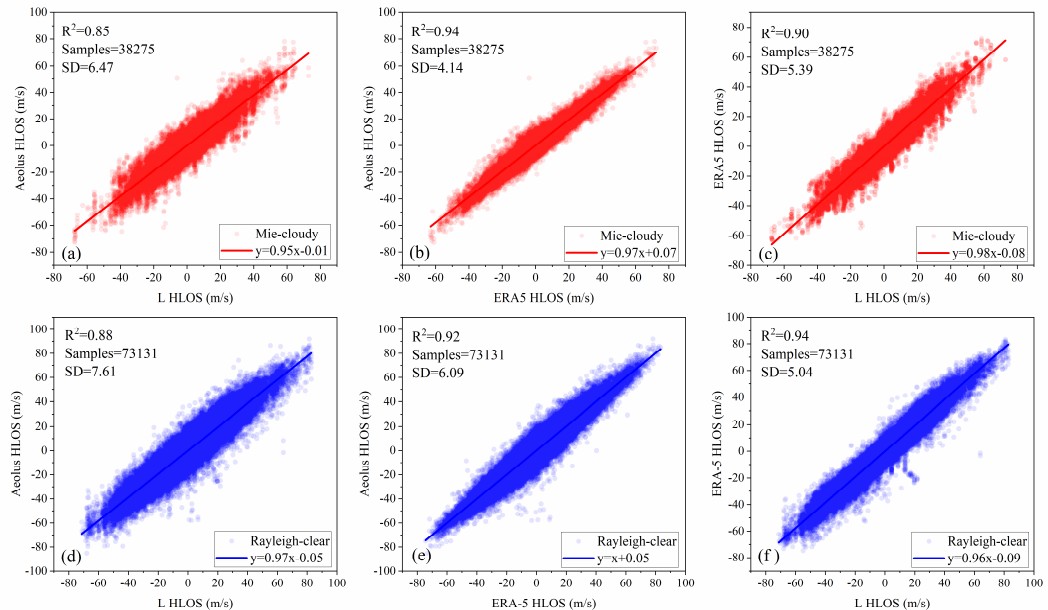

**Figure 4. Comparison results of Aeolus, L-band RS, and ERA5 data in Mie-cloudy and Rayleigh-clear groups. (a, d) Aeolus vs L-**
160 **band RS, (b, e) Aeolus vs ERA5, and (c, f) ERA5 vs L band RS**

| | Aeolus vs L | | Aeolus vs ERA5 | | ERA5 vs L | |
|---|---|---|---|---|---|---|
| | Mie-cloudy | Rayleigh-clear | Mie-cloudy | Rayleigh-clear | Mie-cloudy | Rayleigh-clear |
| R | 0.92 | 0.94 | 0.97 | 0.96 | 0.95 | 0.97 |
| N samples | 38275 | 73131 | 38275 | 73131 | 38275 | 73131 |
| Slop | 0.95 | 0.97 | 0.97 | 1.00 | 0.98 | 0.96 |
| Intercept(m/s) | -0.01 | -0.05 | 0.07 | 0.05 | -0.08 | -0.09 |
| MD(m/s) | -0.03 | -0.03 | 0.05 | 0.05 | 0.21 | 0.24 |
| SD(m/s) | 6.47 | 7.61 | 4.14 | 6.09 | 5.39 | 5.04 |
| Scaled MAD(m/s) | 5.76 | 6.98 | 3.62 | 5.39 | 2.04 | 1.94 |

**Table 2. Comparison between Aeolus, L HLOS and ERA5 HLOS wind**

| | Aeolus vs L | | Aeolus vs ERA5 | | ERA5 vs L | |
|---|---|---|---|---|---|---|
| | Mie-cloudy | Rayleigh-clear | Mie-cloudy | Rayleigh-clear | Mie-cloudy | Rayleigh-clear |
| R | 0.96 | 0.95 | 0.97 | 0.96 | 0.98 | 0.99 |
| N samples | 4482 | 8067 | 4482 | 8067 | 4482 | 8067 |
| Slop | 0.96 | 0.99 | 0.97 | 1.00 | 0.99 | 0.98 |
| Intercept(m/s) | -0.22 | -0.02 | -0.01 | 0.19 | -0.22 | -0.19 |
| MD(m/s) | -0.24 | 2.5E-3 | -0.01 | 0.20 | -0.23 | -0.20 |
| SD(m/s) | 4.73 | 6.18 | 4.19 | 6.10 | 3.09 | 2.97 |
| Scaled MAD(m/s) | 4.11 | 5.42 | 3.55 | 5.26 | 2.54 | 2.30 |

**Table 3. Same as Table 2, but for rectangle of ±1° lat/lon. The distance of 1° lat/lon is approximately 100km in the regions we studied.**

Table 2 shows the comparison results for the three groups of data. The consistency of the L-band RS and Aeolus data was the lowest among the three groups for the Mie-cloudy and Rayleigh-clear groups. The performance of the R and SD values of this group (Aeolus vs L) was also slightly worse than that of the other two groups, which was expected. In addition, the scaled MAD value of group ERA5 vs L was significantly lower than that of other groups. This means that there is a great agreement between ERA5 wind and L-band RS wind, despite their temporal and spatial matching problems. Overall, the correlation coefficients in the three sets of comparison results were all higher than 0.92, reflecting the reliability of the data used in the present study. The comparison results of data 1° from RS site were also shown in Table 3. The correlation coefficient is increased and the scaled MAD is decreased, when applying a stricter collocation criteria of 100 km. Stricter collocation criteria led to better results.

Because the present study also involved wind field data in different regions, the influence of the geographical location and climatic factors on data quality needed to be demonstrated. Figure 5 shows a comparison of the Aeolus and L-band RS data in the four regions used in the present study. The blue data points represent the data of the Rayleigh-clear group and the red data points represent the data of the Mie-cloudy group. The colors of the linear fitting line and related parameters are consistent with the corresponding data points. Table 4 summarizes the results of this comparison. The consistency of the Aeolus and L-band RS data in the Qingyuan area was worse than that of the other three groups, which might be because Qingyuan is close to the tropics, where the atmospheric convection is active. In addition, from the perspective of the correlation coefficient R, the correlation of the Rayleigh-clear group between the Aeolus and the L-band RS data was higher; however, the SD value was also relatively higher than that of the Mie-cloudy group, which means that the data points were more scattered. As the latitude decreased, the data quality declined; however, the data quality of Baoshan is similar to that of Chifeng, which means that this trend was not obvious.

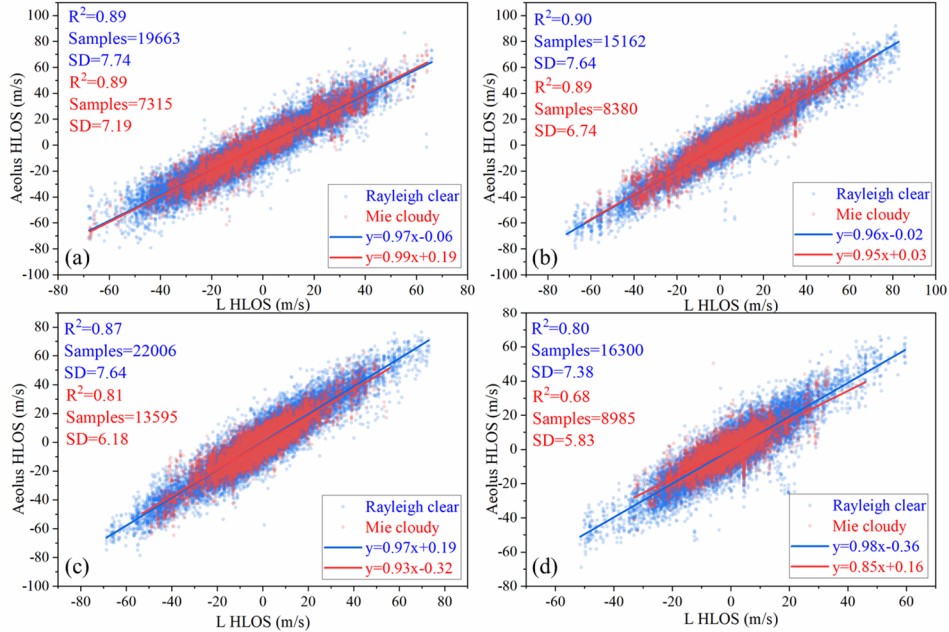

**Figure 5. Comparison of Aeolus and L-band radar detection data in four regions. The blue data points are the Rayleigh-clear group data and the red data points are the Mie-cloudy group data. (a) Chifeng, (b) Baoshan, (c) Shapingba, and (d) Qingyuan.**

| | Chifeng | | Baoshan | | Shapingba | | Qingyuan | |
| --- | --- | --- | --- | --- | --- | --- | --- | --- |
| | Mie-cloudy | Rayleigh-clear | Mie-cloudy | Rayleigh-clear | Mie-cloudy | Rayleigh-clear | Mie-cloudy | Rayleigh-clear |
| R | 0.94 | 0.94 | 0.94 | 0.95 | 0.90 | 0.93 | 0.82 | 0.89 |
| N samples | 7315 | 19663 | 8380 | 15162 | 13595 | 22006 | 8985 | 16300 |
| Slope | 0.99 | 0.97 | 0.95 | 0.96 | 0.93 | 0.97 | 0.85 | 0.98 |
| Intercept (m/s) | 0.19 | -0.06 | 0.03 | -0.02 | -0.32 | 0.19 | 0.16 | -0.36 |
| MD (m/s) | 0.20 | 0.19 | -0.19 | -0.21 | -0.30 | 0.20 | 0.28 | -0.37 |
| SD (m/s) | 7.19 | 7.74 | 6.74 | 7.64 | 6.18 | 7.64 | 5.83 | 7.38 |

**Table 4. Comparison results of Aeolus and L-band RS in different regions**

The data qualities of the ERA5 and L-band RS wind data were further verified as the reference data. Previous studies have shown that ERA5 and RS wind data are relatively reliable in various wind field models and detection methods (Ingleby, 2017; Piasecki et al., 2019; Ramon et al., 2019; Molina et al., 2021). The ERA5 data (matching the RS data) used in the present study were compared and verified with the RS data in the vertical direction, and the results are shown in Figure 6. Except for the vertical height range of 14–19 km, the average error between the RS and ERA5 wind field data was between -0.5 m/s and 0.5 m/s. However, Figure 6b shows that the increase in the error between 14 km and 19 km was caused by the increase in the wind speed value, and the relative error between the RS and the ERA5 wind data was within 0.3. The good consistency of the two datasets in the vertical direction indirectly verified the quality of the ERA5 and RS wind field data in the present study.

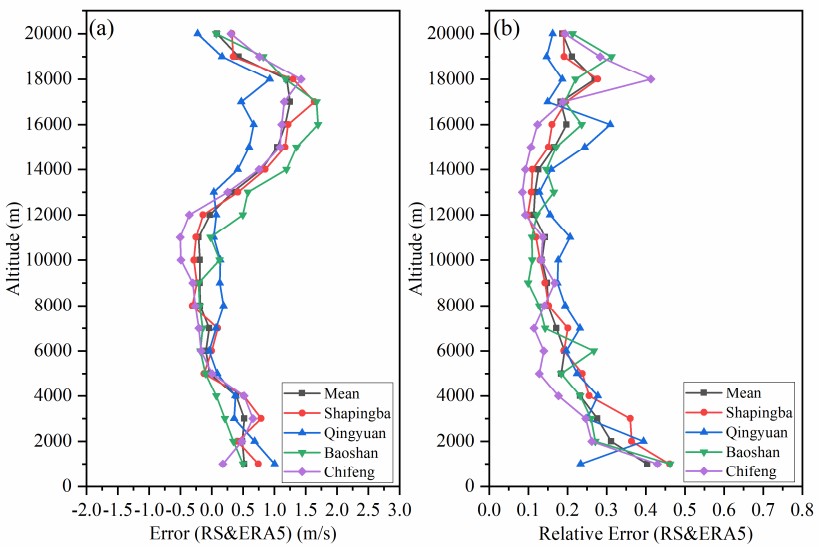

**Figure 6. Vertical comparison of ERA5 and RS wind data for four regions. (a) Error and (b) Relative error**

## 3.2 Seasonal variations in relative errors

After the data quality was confirmed, the three wind field datasets were introduced into Eqs. (3) – (9) to calculate the errors. The representative statistical distribution of errors is shown in Figure 7a, which conforms to the Gaussian distribution law.

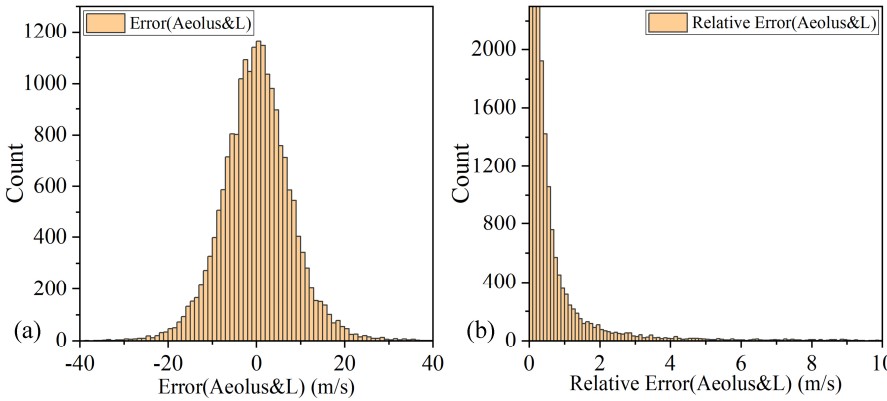

**Figure 7. Statistical distribution of the error (a) and relative error (b) between Aeolus detection and L-band RS data. The data were from the Rayleigh-clear group for 12 months in Chifeng.**

However, the value of the relative error is affected by the denominator in Eq. (7). When the value of the denominator is close to zero, the relative error has a larger outlier. The occurrence of such an outlier was sporadic and random, as shown in Figure 7b. Although most of the relative errors were distributed in the interval [0,3], sporadic outliers still existed in the range greater than 3. These outliers affect the subsequent calculations and must be filtered. The threshold screening method was selected for filtering, and the relative error that was greater than 3 (i.e., 300%) was considered an invalid value. The statistical results of the relative error at the four RS stations are summarized in Table 5. When we choose 3 (300%) as the relative error threshold, the most meaningful data points in the four regions were within the threshold.

|            | P90  | P95  | P99    |
|------------|------|------|--------|
| Chifeng    | 1.80 | 3.91 | 499.24 |
| Baoshan    | 2.45 | 5.24 | 41.75  |
| Shapingba  | 2.96 | 5.92 | 26.55  |
| Qingyuan   | 3.33 | 6.29 | 30.42  |

**Table 5. Percentiles of the relative error between Aeolus detection and L-band RS data for four regions. P90 represents the boundary value of the range where 90% of the relative error falls. The mean of P95 and P99 are similar to P90.**

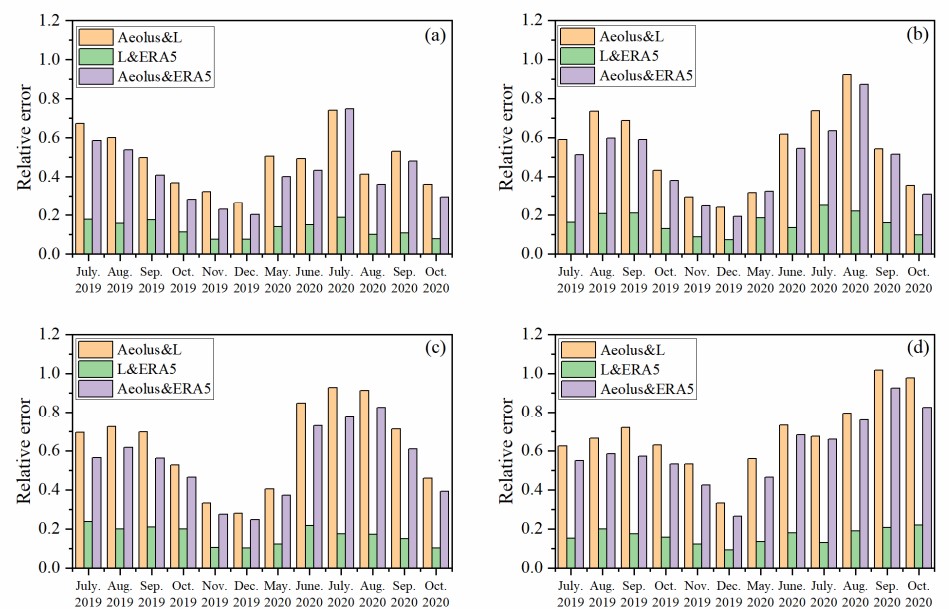

**Figure 8. Monthly mean values of the three groups of relative errors in different regions and months, (a) Chifeng, (b) Baoshan, (c) Shapingba, and (d) Qingyuan.**

Further, we used Eq. (8) to calculate the monthly average relative error for each month in each region. Then, Eq. (9) was used to calculate the monthly average of the Aeolus relative error parameters. Finally, we obtained the changes in the three sets of relative errors (Aeolus&L, L&ERA5, and Aeolus&ERA5) from July 2019 to October 2020 (data missing from January to April 2020), as shown in Figure 8. For the Rayleigh-clear data, the relative error of the Aeolus data was significantly larger in summer. Although the $\overline{D}_r(L\&ERA5)$, which represents the error caused by imperfect space-time matching, also increased in

summer, the increase in $\overline{D}_r(Aeolus\&L)$ and $\overline{D}_r(Aeolus\&ERA5)$ in the summer months was much larger than that of $\overline{D}_r(L\&ERA5)$. This increase in the relative error was caused by the variation in the Aeolus wind product performance. The same seasonal trend in the Rayleigh-clear data in the four regions (Fig. 9a) also confirmed this, as the mean relative error parameter in July was over 174% higher than that in December. The mean random error was also calculated and had a 0.97m/s increase in July than December. We also found that as the latitude decreased, the month in which the relative error peak

appeared was delayed. Over the two years (2019–2020), the peaks of the relative errors for Chifeng were in July, whereas for Baoshan and Shapingba they were in August and Qingyuan was delayed to September. The relative errors in the summer of 2020 were higher to varying degrees than those in 2019, which was mainly caused by the decrease in the output laser energy of Aeolus. This effect is less pronounced for Mie-cloudy winds, because they are not as strongly depended on the laser energy. We also calculated the monthly average value of $D_{Aeolus}$ in the Mie-cloudy group, as shown in Figure 9b. The monthly mean

value of $D_{Aeolus}$ in the Mie-cloudy group did not show a significant seasonal trend, similar to the Rayleigh-clear group. The mean relative error parameter in July was only 39% higher than that in December. Its seasonal fluctuations were relatively random for the different regions; however, the summer relative error was slightly larger overall.

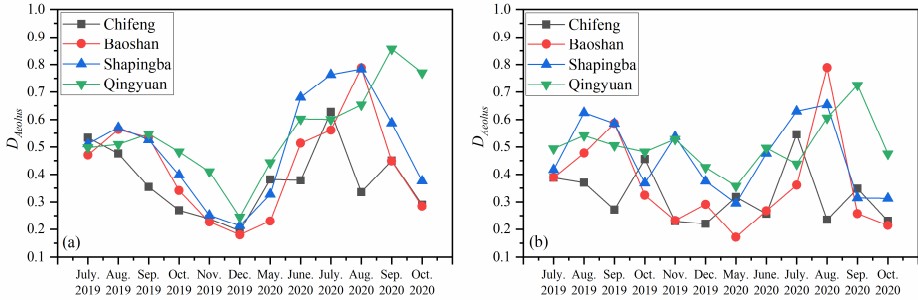

**Figure 9. Changes in the monthly average of $D_{Aeolus}$ in different months, (a) Rayleigh-clear and (b)Mie-cloudy**

The ECMWF proposed that varying temperature gradients across the instrument's mirror M1 caused seasonal fluctuations in the quality of the Aeolus detection data (Rennie and Isaksen, 2020). After applying corrections for the mirror effects, the seasonal variation caused by the thermal structure of the system itself theoretically became very small. Seasonal variations in relative errors may be partly due to seasonal variations in atmospheric conditions.

Considering the difference in the detection range between the Rayleigh-clear and Mie-cloudy groups (Mie-cloudy mainly
detects the aerosol layer), this seasonal difference might be caused by seasonal changes in the real atmospheric environment. Therefore, we proposed and verified two conjectures based on the Aeolus working principle.

### 3.2.1 Seasonal variations in atmospheric wind direction

Because Aeolus detects only a single line-of-sight wind vector, the detection wind vector is a component of the real wind vector. When the angle between the detection and real wind vectors approaches 90°, the real wind vector contributes almost
nothing to the detection wind vector and the detection error is the largest. The closer the angle between the two is to 0° or 180°, the smaller is the detection error.

The wind direction of the atmospheric wind field exhibited an obvious seasonal trend. In China, the northwest monsoon prevails in the winter and the southeast in the summer (Chang, 2004; HUANG et al., 2004; He et al., 2007). To verify whether the seasonal variation in wind direction was the cause of the seasonal variation in the Aeolus wind product performance, we
analyzed the statistical distribution of the angle between the real horizontal wind and the Aeolus HLOS direction.

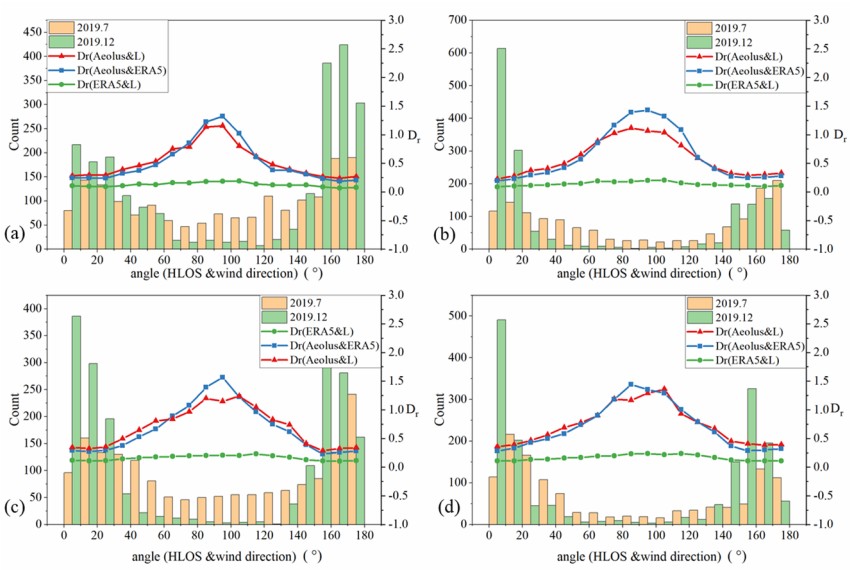

**Figure 10. Statistical distribution in different angle (HLOS and wind direction) about the data points and mean relative errors from the Rayleigh-clear group data. The relative error Dr in the figure was calculated from the 12-month data of the region; (a) Chifeng, (b) Baoshan, (c) Shapingba, and (d) Qingyuan.**

Based on the previous data matching work, we calculated the angle α between the real horizontal wind direction (provided by the ERA5 data) and the Aeolus HLOS direction (provided by the L2B data) of each Aeolus valid data point. Figure 10 was

obtained from the Rayleigh-clear group data. When the angle α was between 70° and 110°, the relative error of the Aeolus data increased significantly. The proportion of data points with angles between 70° and 110° in July was 8.14% higher than that in December. Most of the data points in December were concentrated in the vicinity of 0° and 180°. Theoretically, this

would significantly increase the average relative error of Aeolus in July, which might be one of the reasons for the increase in the relative error of Aeolus during the summer. For the Mie-cloudy group (Fig. 11), the proportion of data points with angles between 70° and 110° in July was 5.86% higher than that in December; therefore, there was the same order of magnitude for the angle difference distribution of the Rayleigh-clear and Mie-cloudy data. However, the seasonal variation of the relative error in the Mie-cloudy group was much smaller than that in the Rayleigh-clear group; therefore, this conjecture cannot explain

the seasonal performance of the Mie-cloudy group.

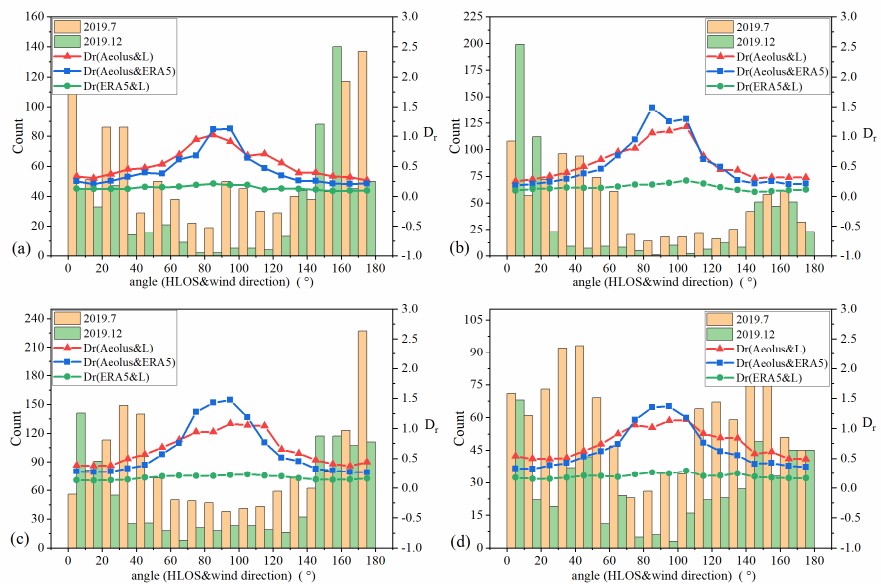

**Figure 11. Statistical distribution in different angle (HLOS and wind direction) about the data points and mean relative errors from the Mie-cloudy group data. The relative error Dr in the figure was calculated from the 12-month data of the region; (a) Chifeng, (b) Baoshan, (c) Shapingba, and (d) Qingyuan.**

**3.2.2 Seasonal variations in upper cloud cover**

During the actual work process, the spaceborne wind lidar is susceptible to the influence of cloud aerosols. When the laser passes through the cloud aerosol layer, it is subjected to a strong attenuation effect, resulting in a decrease in the energy of the laser beam and signal-to-noise ratio (SNR). Simultaneously, for the Rayleigh channel, the cloud aerosol layer will cause strong Mie scattering, which will pollute the signal of the Rayleigh channel and increase its detection error (Rennie et al., 2020).

Because the Rayleigh channel has comprehensive coverage in altitude, we took the effective data of the Rayleigh channel (both Rayleigh-clear and Rayleigh-cloudy) and counted the number of samples ($N_{clear}$, $N_{cloudy}$) based on the month and altitude, and then defined the parameter $r$ that represented the backscatter ratio:

$$r = \frac{N_{clear} + N_{cloudy}}{N_{clear}}, \tag{10}$$

$r$ was normalized to obtain the parameter $R$. The larger the value of the parameter $R$, the stronger the scattering of Mie in the

altitude layer. We calculated the $R$ value of 12 months in different regions with a height resolution of 1000 m to obtain Figure 12. Obvious high-altitude Mie scattering layers existed for the four areas during summer, whereas during autumn and winter, this Mie scattering layer moved to areas with lower altitudes, with the Mie scattering intensity weakened. In Qingyuan, the high-altitude Mie scattering layer in November and December 2019 was very weak (Fig. 12d).

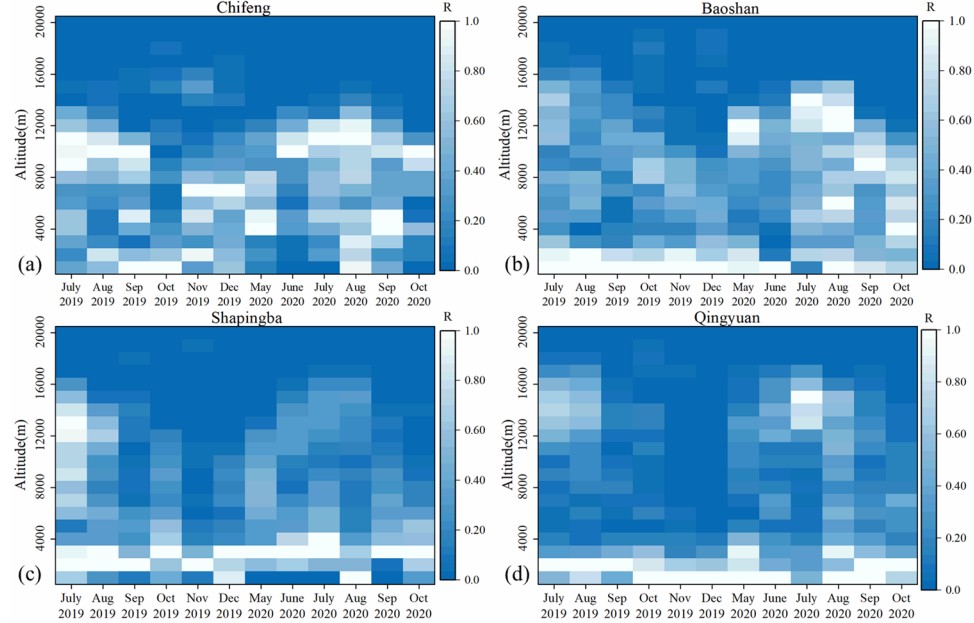

**Figure 12. Vertical distribution of R values in different regions and months. (a) Chifeng, (b) Baoshan, (c) Shapingba, and (d) Qingyuan**

The ERA5 cloud coverage information matching with the Aeolus data points are shown in Figure 13. The $r$ value in Figure 13 represents the backscattering ratio, defined by Eq. (10). The mean cloud coverage and $r$ values had a similar trend in the vertical direction; however, they were different in the near-ground area because the main reason for the Mie scattering here was aerosols rather than clouds. Taken together, the high-altitude Mie scattering layer in Figure 12 in summer was caused by

the presence of clouds in the area.

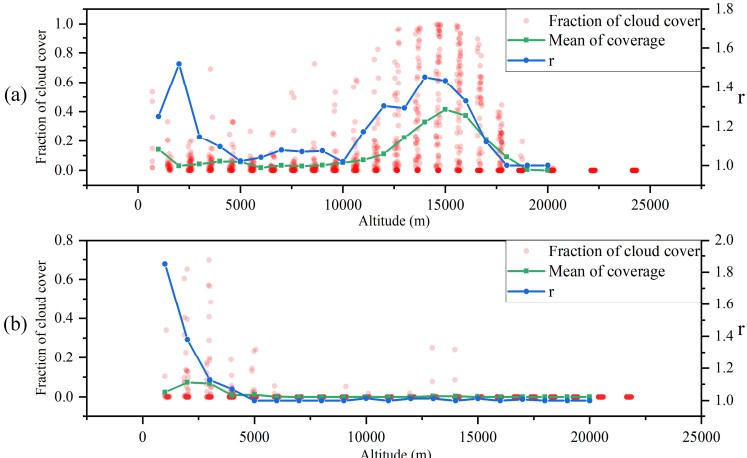

**Figure 13. ERA5 cloud coverage of the valid Rayleigh channel data in the vertical direction. Data in (a) is from Qingyuan in July 2019, and in (b) is from Qingyuan in December 2019. The green line is the mean value of cloud coverage at different distance gates.**

Therefore, the height of the cloud tops in summer increased significantly within the Aeolus detection height range. Combining the results (Fig. 12 and Fig. 13), the cloud top height in July was approximately 3–5 km higher than that in December for the four regions, which was consistent with the seasonal variation rule of cloud top height in East Asia (Zhao et al., 2020).

When the satellite-borne wind lidar worked, the high-altitude cloud layer attenuated the laser beam and reduces

the energy of the laser beam passing through the cloud layer. Thus, the SNR of the echo signal in the area below the high cloud decreased, and the detection error increased. Simultaneously, the Rayleigh signal from the cloudy area was interfered with by the Mie scattering signal, which affected the calculation of its Doppler shift. Although the Aeolus data processing algorithm uses a strict backscatter ratio threshold to remove the Rayleigh channel

data elements that may contain Mie scattering from the Rayleigh-clear group (Rennie et al., 2020), the signal interference still affects the final inversion result.

Finally, we attempted to use this conjecture to explain why the relative error seasonal variation of the Mie-cloudy group was not obvious (Fig. 9b). There are two special data points in Figure 9b: July and August 2020 in Baoshan. These were both during the summer; however, the average relative error in August was significantly higher than that in July. The vertical distribution of the valid data points and all data points of the Mie-cloudy group in the two months are shown in Figure 14. The data points in July were mainly distributed in the high-altitude cloud area, which might be due to the dense clouds in the upper air in July making it difficult for the Mie channel to detect the area below the clouds. The Mie scattering signal of the high-altitude cloud layer close to the satellite, i.e., small optical thickness and high SNR, improved the Aeolus data quality. In addition, the Mie channel discriminator was not sensitive to the Rayleigh scattering signal; therefore, it was unnecessary to consider the Rayleigh signal interference (Rennie et al., 2020). In August, the data points were evenly distributed in the vertical direction, and the amount of low-altitude data accounted for a considerable proportion. Before the laser beam reached a low altitude, it was attenuated by high-altitude clouds. The low-altitude Mie scattered echo signal generated by the laser beam propagated through the upper atmosphere and attenuated again. Therefore, it became very weak when it reached the Aeolus receiving telescope, which was not conducive to subsequent signal processing and reduced the quality of the Mie-cloudy data. $D_{Aeolus}$ was significantly higher in August than in July when the altitude was lower than 8 km, which supported the above explanation (Fig. 14). Therefore, in summer, a considerable part of the data in the Mie-cloudy group came from a high altitude; however, the low-altitude and the high-altitude Mie scattering signals contributed to the overall relative error in the opposite way, which might be why the seasonal variation of the Aeolus relative error in the Mie-cloudy group was not obvious.

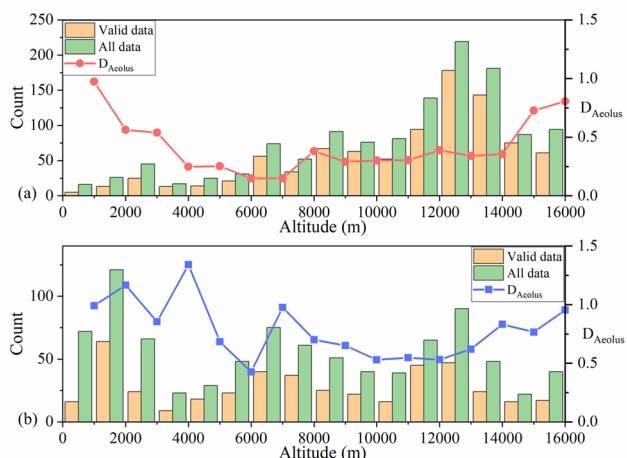

**Figure 14. Distribution of Mie-cloudy data points and $D_{Aeolus}$ in altitude. (a) July 2020, and (b) August 2020. Data are from Baoshan.**

## 4 Conclusion

In the present study, the seasonal variation in the Aeolus wind product performance in China was analyzed using Aeolus detection data, L-band RS detection data, and ERA5 data from July to December 2019 and May to October 2020. First, the difference between the Aeolus and L-band RS data was discussed, and the selection threshold of the Aeolus data estimation error was clarified, which was 8 m/s for the Rayleigh-clear data and 4 m/s for the Mie-cloudy data. After the valid data were filtered, a comparative analysis of the three wind field datasets was undertaken. The R value of Aeolus's Rayleigh-clear (Mie-cloudy) data and ERA5 data was 0.96 (0.97), and the R value of the L-band RS data was 0.92 (0.94). Therefore, the Aeolus detection data were in good agreement with the ERA5 and L-band RS data, and the quality of the data used in the present study was reliable.

These three datasets were then used to calculate the monthly mean value of the relative error. The calculation results showed

that the relative error of the summer Rayleigh-clear data in the four regions increased significantly, and the average relative error in July was 174% higher than that in December. In the Mie-cloudy group, this seasonal trend was not obvious, and the performance was more random. Combining the working principle of Aeolus, we proposed two conjectures to explain the seasonal variation in the relative error of the Aeolus data. One was the variation in the angle between the actual horizontal wind direction and Aeolus HLOS direction, which might have affected the extent of the Aeolus single vector data to reflect

the true wind vector. The other was the seasonal variation in the altitude of the high-altitude clouds and cloud tops, which might have affected the SNR of the echo signals in different channels.

For the first conjecture, we calculated the distribution of the angle between the actual horizontal wind direction and the Aeolus HLOS direction of different regions in July and December. There were more data points distributed in the high error interval of 70°–110° in July than in December; therefore, this conjecture was reasonable. However, the first conjecture encountered

problems in explaining the situation of the Mie-cloudy data. For the second conjecture, we set the parameter $R$ to represent the backscattering ratio and calculated the distribution of $R$ at different altitudes in different months. There was a strong high-altitude Mie scattering layer in summer. Combined with the cloud coverage information, the Mie scattering layer was caused by the high-altitude clouds in summer. The high-altitude clouds reduced the SNR of the echo signal received by Aeolus and interfered with the signal analysis and processing of the Rayleigh channel. This conjecture is also reasonable and can explain

the seasonal variation in the relative error of the Mie-cloudy channel Aeolus

In the present study, the analysis of the Aeolus data quality and its seasonal changes in four regions (Chifeng, Baoshan, Shapingba, and Qingyuan) of China will help to better understand and use the Aeolus detection data over China. Besides, this study is helpful to figure out the influence of clouds and wind direction on the detection performance of Aeolus, which will provide a reference for the follow-up development of spaceborne wind lidar.

**Data availability**

The Aeolus L2B data used in this study can be downloaded from https://aeolus-ds.eo.esa.int/oads/access/collection (last accessed March 13, 2021). The L-band RS data can be provided for non-commercial research purposes upon request (Yinchao Zhang: ychang@bit.edu.cn). The ERA5 data can be downloaded from https://cds.climate.copernicus.eu/cdsapp#!/dataset/ 10.24381/cds.bd0915c6?tab=overview (last accessed March 13, 2021).

**Author contributions**

S. Chen and R. Cao conceived the idea for verifying the seasonal variation of Aeolus wind product performance over China; R. Cao and Y. Xie conducted the data analyses and co-wrote the manuscript; Y. Zhang, W. Tan, H. Chen, P. Guo, and P. Zhao discussed the experimental results. All coauthors have helped in reviewing the manuscript.

**Competing interests**

The authors declare that they have no conflict of interest.

**Acknowledgements**

We are very grateful to the China Meteorological Administration for the L-band RS data in China, the ESA for the Aeolus data, and the ECMWF for ERA5.

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
