# Peer review of "Study of the seasonal variation of Aeolus wind product performance over China using ERA5 and radiosonde data"

_Atmospheric Chemistry and Physics, 2021_

## Referee Comment (RC2)

**Anonymous reviewer #2:**

An assessment of the Aeolus L2B Rayleigh-clear and Mie-cloudy wind product quality is presented by this study, using an impressive network of L-band radiosonde stations over China, complemented by ERA5 data. The study focuses on the seasonal variations of the Aeolus wind product quality, including a comprehensive analysis of the impact of wind direction and presence of clouds in different altitude levels. The presented results are valuable for further analysis of satellite products over China, including aerosol and air quality studies. The manuscript is well written, but some major revisions are needed before publication, which are described in the following:

1. In my opinion, the phrase 'Aeolus detection performance' is a bit misleading. I would expect that this is used for minimum detectable aerosol layers or more the limitations of the instrument. But you are analysis the wind product performance. I would advise to avoid this phrase and rather use 'Aeolus wind product performance' or 'Aeolus wind data quality'.

2. You define the relative error as representative for the detection performance (product quality…). Other studies, like Martin et al. 2021, Baars et al., 2020, Witschas et al. 2020, Lux et al. 2020 all use the scaled Median Absolute Deviation (MAD) as a measure of the product quality, which should be closest to the random error of Aeolus winds, compared to the ECMWF model (as done in NWP monitoring). Your results would fit better in the selection of Aeolus validation studies, if you would also show the scaled MAD.

3. You applied a collocation distance of 2.5 deg lat/lon for valid overpasses. However, the cal/val implemtation plan (https://earth.esa.int/eogateway/documents/20142/1564626/Aeolus-Scientific-CAL-VAL-Implementation-Plan.pdf) suggests to use a horizontal collocation distance of 100 km around the ground site, which is much stricter than 2.5 deg. I understand that depending on the wind direction, the radiosonde might drift closer towards the actual Aeolus wind profile, but some further justifications of the 2.5 deg criteria would be desirable. Particularly for Mie-cloudy winds you can have significant deviations when using such a large distance.

**Detailed comments:**

*Line 15: Relative error 174% higher than…*
> I would recommend to mention mean random wind errors of Aeolus against L-band radiosonde data. For external readers, a random error in m/s is more easily understandable than a %.

*Line 25: which uses a single-view detection method to scan the global three- dimensional wind field from space…*
> ALADIN does not scan the 3D wind field, just the Horizontal line-of-sight (HLOS) component of the 3D wind field. Please rephrase.

*Line 32: Level 2B products provide HLOS wind data after actual atmospheric correction and deviation correction*

> Better: fully processed HLOS wind profiles after correction of temperature and pressure effects.

*Line 34: relevant researchers*

> dedicated calibration and validation team have carried out …

*Line 48: After the implementation of the new M1 deviation correction scheme, the effect of system thermal performance changes on Aeolus' seasonal fluctuation is theoretically excluded.*

> *… is* significantly reduced which led to systematic errors lower than 1 m/s (Rennie and Isaksen, 2020)

*Line 55: Data and methods:*

> I would recommend to rearrange the chapter just a little bit. Start with Aeolus L2B, L-band, ERA5 and put the description of the collocation criteria together with Fig. 1 under 2.4, when you explain the data matching.

Line 76: What is the wind uncertainty of the L-band data. Please add a reference for the radiosonde wind bias here.

Line 85: You mention that ERA5 data is often used as reference in meteorological data analysis. However, we know that modelled wind predictions have high uncertainties too, particularly in cloudy situations. Please add some information of the expected model wind uncertainties in this chapter.

*Line 99: Conversion from ERA5 to Aeolus HLOS:*

> In Eq. 1, you use the wind direction and horizontal wind speed of the radiosonde data and the Aeolus azimuth angle to reproject the RS data to Aeolus data. I don't understand, why you do the same with the ERA5 data, where you have the full u and v wind vector information.
> Wouldn't be the correct conversion: HLOS = $-u \cdot \sin(\phi) - v \cdot \cos(\phi)$ ?

*Line 115*: Please spell check the variable V_ture, it should be V_true I guess?

*Line 124*: You define the relative error as representative for the detection performance (product quality…). Other studies, like Martin et al. 2021, Baars et al., 2020, Witschas et al. 2020, Lux et al. 2020 all use the scaled Median Absolute Deviation (MAD) as a measure of the product quality, which should be closest to the random error of Aeolus winds, compared to the ECMWF model (as done in NWP monitoring). Your results would fit better in the selection of Aeolus validation studies, if you would also show a scaled MAD.

*Fig. 3*: It is recommended, also during several Aeolus validation workshops, to have HLOS Aeolus on the y-axis and all reference observations, models on the x-axis. Please swop Aeolus and L-band on the fig.3 (a,d). I would further advise to have Aeolus-L-band first,

Aeolus-ERA5 in the middle, and ERA5-L band as last plot. You are describing (a,b…) but the letters are not shown in the plot.

*Lines 146-153:* You are mentioning the space time matching problem. In the Cal/Val implementation plan, it is recommended to use a collocation criteria of +-100 km (which is much less than 2.5 deg) and +-60 min in time. One possibility could be to show the results for all measurements and one table for all within the 100 km distance. This may improve your results.

*Fig. 5:* In order to compare it easier to the random error evolution, presented by Rennie and Isaksen, 2020, I would recommend to show the scaled MAD, additionally to the relative error. As mentioned before, other studies one the scaled MAD as the main estimate of the wind product random error.

*Lines 194-196:* The fact that the relative errors are larger in summer 2020 compared to summer 2019 is mainly caused by the decrease in the output laser energy of ALADIN. This effect is less pronounced for Mie-cloudy winds, because they are not as strongly depended on the laser energy.

*Lines 205-210: On the other hand, the different situation of the Mie-cloudy group and the Rayleigh-clear group does not seem to support the explanation that the Aeolus system itself causes seasonal changes in detection performance…*

> What do you mean with that? Please add some further explanations on that. NWP monitoring results and other independent validation studies show that the Mie-cloudy bias is much less effected by thermal variations compared to the Rayleigh-clear bias. This can be explained by technical differences of the Mie spectrometer compared to the Rayleigh spectrometer.

*Line 216: The wind direction of the atmospheric wind field has an obvious seasonal trend.*

> Can you provide a reference publication for this? I believe there are some manuscripts showing the seasonal trends of the wind direction in China.

Line 224-225: Based on the previous data matching work, we calculate the angle α between the real horizontal wind direction (provided by 225 ERA5 data) and the Aeolus HLOS direction (provided by L2B data) of each Aeolus valid data point…

> How do you calculate the wind direction from the HLOS data? Or do you mean that you use the azimuth angle, given in the L2B product?

Line 240: …*resulting in decrease in the energy of the laser beam and signal-to-noise ratio (SNR)*:

> Instead of decrease in energy, better say: "resulting in a decrease of the laser return signal and…"

*Line 242: we takes the effective data* → we take the effective data…

Line 256: Mie-cloudy data does not distinguish between aerosol and clouds. You mention the impact of aerosols here, but some further comments about the potential impact of low-level aerosol layers on the wind product quality would be desirable. Could seasonal variations of aerosol layers also influence the results?

*Fig. 12:* Please ad (a) and (b) to the image.

Line 319: Avoid to use double reasonable.

*311-320:* Have you thought about the effect of horizontal cloud variability on the Mie-cloudy wind errors? High clouds tend to be more homogeneous than low level clouds, which can vary strongly within one ~10 km Mie-cloudy wind result. This could potentially cause the higher relative errors in lower altitudes. Furthermore, the lidar signal is usually attenuated, if there are high clouds, thus I don't think that there will be Mie-cloudy valid winds from low altitude levels below high-altitude Mie-cloud valid winds.

*Line 323: Besides, as the first spaceborne wind lidar, the analysis of factors affecting the detection performance of Aeolus will help provide a reference for the follow-up development of spaceborne wind lidar.*
> Could you be a bit more specific what of your findings will contribute to the follow-on development of future spaceborne wind lidars? Just add some more details here.

---

## Author Response (AR1)

**Author's Response to Referee #1**

In this response, the referee comments (in black) are listed together with our replies (in blue) and the changes to the original manuscript (in red).

In this study, the Aeolus L2B wind product quality is verified by ERA5 and L-band radiosonde (RS) data. The accuracy of the Aeolus Rayleigh-clear data and Mie-cloudy data in four regions and different seasons in China and the reasons for the errors are explored by accounting for various potential factors. The evaluation method is generally sound and the results of error analyses are highly valuable for using new satellite product for various applications in the region. However, the paper needs significant improvements before it can be considered for publication.

**Responds:** Thanks a lot for your reviews on our manuscript entitled "Study on the seasonal variation of Aeolus detection performance over China using ERA5 and radiosonde data". The comments are valuable and very helpful for revising and improving our paper. We have revised the manuscript according to the comments and the details are shown as follows.

1. As the main objective of the study is to assess the validity and uncertainty of the Aeolus retrieval using the ERA5 and RS data, the quality and uncertainty of the latter two products must be well documented. While ERA5 data have widely used, its accuracy varies considerably, pending on the availability and usage of observation data used in the data assimilation system. Were ERA5 data accurate everywhere, there would be no need to launch the Aeolus.

**Responds:** Suggestions accepted. We have supplemented the quality verification of ERA5 data and RS data used in the manuscript (see Section 3.1). Although the uncertainty of ERA5 data varies pending on the availability and usage of observation data, the verification results show that the ERA5 data of the four regions used have high data quality. Therefore, this study uses ERA5 data as a reference to verify that the seasonal variation of wind and cloud detection efficiency is reasonable. The validity statement about RS data is added in Section 2.2, L78:

L78:

Added, "The L-band RS wind data used in this study are all the valid data detected by the four L-band RS stations."

More information about the quality of ERA5 wind data and RS wind data used is supplemented at the end of Section 3.1, L167.

L167:

Added, "The data qualities of the ERA5 wind data and L-band RS wind data are further verified as they are reference data. Studies have shown that ERA5 wind data and RS wind data are relatively reliable in various wind field models and detection methods (Molina et al., 2021; Piasecki et al., 2019; Ramon et al., 2019; Ingleby, 2017). The ERA5 data (matching the RS data) used in this study was compared and verified with the RS data in the vertical direction, and the results are shown in Figure 6. Except for the vertical height range of 14–19 km, the average error between RS wind data and ERA5 wind field data is basically within the range of -0.5–0.5m/s. However, it can be seen from Fig. 6b that the increase in the error within the range of 14–19 km is caused by the increase in the wind speed value, and the relative error between the RS wind data and the ERA5 wind data is mostly within 0.3. The good consistency of the two data in the vertical direction can indirectly verify the

quality of the ERA5 wind field data and the RS wind field data in this study."

[Figure]

**Figure 6. Vertical comparison of ERA5 wind data and RS wind data for four regions. (a) Error, (b) Relative Error.**

2. The method needs to be described more clearly in terms of logical expression and working principles.

**Responds:** Suggestions accepted. More information in terms of logical expression and working principles has been added in the Section 2, L59:

L59:

Added, "The main processing procedures is showed in Fig. 2 and more details will be introduced below."

Added a figure below Fig.1:

[Figure]

**Figure 2. The main processing procedures of Aeolus, L-band RS and ERA5 wind data.**

3. The paper needs to be edited extensively to correct many non-standard use of English such as "L34: relevant researchers…", "L42: deepened our understanding".

**Responds:** Suggestions accepted. The manuscript has been thoroughly revised by Elsevier language editing services. The certificate is displayed here.

[Figure]

Certificate of Elsevier Language Editing Services

The following article was edited by Elsevier Language Editing Services:
"Study on the seasonal variation of Aeolus detection performance over China using ERA5 and radiosonde data"

Authored by:
Rongzheng Cao

Date: 31-May-2021
Serial number: LE-213079-4AAE7FA25450

4. In Figure 2, add the number of points within and outside the thresholds.

**Responds:** Suggestions accepted. The number of points within and outside the thresholds has been added in Figure 2.

Replace Figure2:

[Figure]

**Figure 2. Difference between the Aeolus HLOS and L HLOS wind components as a function of estimated errors for (a) Mie-cloudy, (b) Mie-clear, (c) Rayleigh-cloudy and (d) Rayleigh-clear. Samples mean the number of data points and the samples blow or beyond the threshold are also listed. Reference lines are the screening threshold of estimated errors: 4 m/s (Mie) and 8 m/s (Rayleigh).**

5. Add a table of all data used in the study.

**Responds:** Suggestions accepted. The table of all data used in the study is added after Figure 1. Meanwhile, a labeling error of Chifeng station has been modified in Figure 1 as follows.

Replace Figure 1:

[Figure]

**Figure 1. The geographic location of the L-band RS site and the Aeolus measurement trajectory. In the partial enlarged view (http://aeolus-ds.eo.esa.int/socat/L1B_L2_Products), the red rectangles represent the range of ±2.5° around the L-band RS stations, and the gray-green lines represent Aeolus' overlapping trajectories of the 12 months.**

Added a table,

| Data | Aeolus | ERA5 | L-band RS |
|------|--------|------|-----------|
| Time | 2019.07–2019.12 and 2020.05–2020.10 Around 10:00 UTC and 22:00UTC | 2019.07–2019.12 and 2020.05–2020.10 Every hour | 2019.07–2019.12 and 2020.05–2020.10 Around 0:00 UTC and 12:00 UTC |
| Geographical location | ±2.5° (latitude and longitude) near the target RS station | ±2.5° (latitude and longitude) near the target RS station | Chifeng(42.3°N,118.8°E) Baoshan(31.4°N,121.4°E) Shapingba(29.6°N,106.4°E) Qingyuan(23.7°N,113.1°E) |
| Vertical Resolution | 0.25km to 2km | 37 pressure levels for 1000hPa to 1hPa | < 10m |
| Max. Altitude | Around 20km for valid data | Around 45km | Around 32km |

**Table 1. Main characteristics of the data sets used in the comparison.**

6. In section 2.4 (L105-110), add some related literature or a further explanation on the wind speed matching method of different data sets in the vertical direction.

**Responds:** Suggestions accepted. Further explanation on the wind speed matching method of different data sets in the vertical direction has been added and amended in Section 2.4, L106.

Section 2.4, L106:

"In the vertical direction, due to the difference in the vertical resolution of the three data, both L-band RS data and ERA5 data are matched with the Aeolus data through linear interpolation. For each Aeolus valid data point, L-band RS (ERA5) data points which are just higher and just lower than the Aeolus valid data point are found. We mark them as $V(H^+)$ and $V(H^-)$ and then do a linear interpolation", changed to,

"In the vertical direction, due to the difference in the vertical resolution of the three data sets, both L-band RS data and ERA5 data need to be matched with the Aeolus data through linear interpolation. Linear interpolation is a method of curve fitting using linear polynomials to construct new data points within the range of a discrete set of known data points. The L-band RS (ERA5) data point which is just higher than one Aeolus data point in altitude is found and marked as $V(H^+)$. Then, $V(H^-)$ is also find which is just lower than the Aeolus data point. The L-band RS (ERA5) wind matched with the Aeolus data point is calculated by Eq. (2)."

We find that there is an error in equation (2), which may bring trouble to readers' understanding. We have corrected Eq. (2) in the manuscript.

Equation (2) changed to:

$$V_H = V(H^-) + \frac{H_{Aeolus} - H^-}{H^+ - H^-} \cdot \left(V(H^+) - V(H^-)\right), \tag{2}$$

7. In L116, how is the data represented by Vture corrected? Are there corrections for ERA5, Aeolus and RS data? The correction process needs further explanation. Does Vture on L125 refer to the same variable as that on line 116? If it is different, use a different expression.

**Responds:** Suggestions accepted. Equation (3) is used to calculate the difference between wind speeds of different data sets and the $V_{ture}$ is not used. The $V_{ture}$ in L125 refer to different variable as that on L116. We have modified the Eq. (3).

Equation (3) changed to,

$$D = V_a - V_b, \tag{3}$$

L116:

"where $V$ represents the detection value, and $V_{ture}$ represents the true value of the calibration", changed to, "where $V_a$ and $V_b$ represent the wind speeds of different data sets."

8. Figure 5 shows the statistical results of the Chifeng station. Are the statistical results at other stations similar to this one? It is recommended to add a description of the table.

**Responds:** Suggestions accepted. Yes. The statistical results at other stations is similar to this one. The statistical results of relative error at other stations is added in Section 3.2, L176.

L177:

Added, "The statistical results of relative error at four RS stations are summarized in Table 3. When we choose 3 (300%) as threshold of relative error, most of meaningful data points in four regions are within the threshold."

|  | P90 | P95 | P99 |
|---|---|---|---|
| Chifeng | 1.80 | 3.91 | 499.24 |
| Baoshan | 2.45 | 5.24 | 41.75 |
| Shapingba | 2.96 | 5.92 | 26.55 |
| Qingyuan | 3.33 | 6.29 | 30.42 |

**Table 3. Percentiles of the relative error between Aeolus detection data and L-band RS data for 4 regions. P90 represents the boundary value of the range where 90% of the relative error falls. The mean of P95 and P99 is similar to P90.**

9. In L195, specify the time interval of the data with low laser energy.

**Responds:** The time interval of the data with low laser energy can't be found in the literature. But

it is reported that the ESA decided to switch to Aeolus' backup lasers in July 2019 due to a drop in laser energy. The Aeolus data used in this paper are from July to December 2019 as well as May to October 2020. Figure 7a showed that relative error in 2020 is apparently higher than 2019, so we suspect that the energy of the new laser is still declining.

10. In L256, the difference in r near the ground is caused by aerosols, but in eq 9 there are only clear and cloudy. Is the type of aerosol also classified into cloudy? Please clarify this.
**Responds:** Suggestions accepted. Yes. The type of scattering caused by aerosol is classified into cloudy group, which has been explained in Aeolus' data processing algorithm (Rennie et al., 2020). We are sorry for not fully clarified it, and now it is emphasized in Section 3.2.2, L246.
L246:
Added, "Scattering caused by clouds and aerosols are both classified into the cloudy group as they belong to Mie scattering."

11. L286, Daeolus represents the relative error value of the Aeolus data. There are many reasons for the error, SNR is one of them. Why can Daeolus represent SNR? The definitions of these two parameters are completely different.
**Responds:** Suggestions accepted. This statement is loosely, we have deleted it in the manuscript. However, the increase in $D_{Aeolus}$ below 8km may be caused by the distribution of clouds. This provides a basis for explaining the insignificant seasonal variation of the detection accuracy of the Mie channel.
L286:
Delete this sentence, "When we use $D_{Aeolus}$ to represent the value of SNR".

12. L304: increases with what?
**Responds:** Suggestions accepted. There may be a misunderstanding here because of the English expression. We have modified it in L304.
L304:
"the relative errors of the Rayleigh-clear data increase significantly for the four regions in the summer as the mean relative error parameter in July is 174% higher than that in December", changed to,
"the calculation results show that the relative error of summer Rayleigh-clear data in the four regions increases significantly, and the average relative error in July is 174% higher than that in December"

**References:**

Ingleby, B.: An assessment of different radiosonde types 2015/2016, European Centre for Medium Range Weather Forecasts, 2017.

Molina, M. O., Gutiérrez, C., and Sánchez, E.: Comparison of ERA5 surface wind speed climatologies over Europe with observations from the HADISD dataset, Int J Climatol, joc.7103, https://doi.org/10.1002/joc.7103, 2021.

Piasecki, A., Jurasz, J., and Kies, A.: Measurements and reanalysis data on wind speed and solar irradiation from energy generation perspectives at several locations in Poland, SN Appl. Sci., 1, 865, https://doi.org/10.1007/s42452-019-0897-2, 2019.

Ramon, J., Lledó, L., Torralba, V., Soret, A., and Doblas-Reyes, F. J.: What global reanalysis best represents near-surface winds?, 145, 3236–3251, https://doi.org/10.1002/qj.3616, 2019.

Rennie, M., Tan, D., Andersson, E., Poli, P., and Dabas, A.: (MATHEMATICAL DESCRIPTION OF THE AEOLUS LEVEL-2B PROCESSOR), 124, 2020.

**Author's Response to Referee #2**

In this response, the referee comments (in black) are listed together with our replies (in blue) and the changes to the original manuscript (in red).

An assessment of the Aeolus L2B Rayleigh-clear and Mie-cloudy wind product quality is presented by this study, using an impressive network of L-band radiosonde stations over China, complemented by ERA5 data. The study focuses on the seasonal variations of the Aeolus wind product quality, including a comprehensive analysis of the impact of wind direction and presence of clouds in different altitude levels. The presented results are valuable for further analysis of satellite products over China, including aerosol and air quality studies. The manuscript is well written, but some major revisions are needed before publication, which are described in the following:

**Responds:** Thanks a lot for your reviews on our manuscript entitled "Study on the seasonal variation of Aeolus detection performance over China using ERA5 and radiosonde data". The comments are very valuable and enlightening, and very helpful for revising and improving our paper. We have revised the manuscript according to the comments and the details are shown as follows.

1. In my opinion, the phrase 'Aeolus detection performance' is a bit misleading. I would expect that this is used for minimum detectable aerosol layers or more the limitations of the instrument. But you are analysis the wind product performance. I would advise to avoid this phrase and rather use 'Aeolus wind product performance' or 'Aeolus wind data quality'.

**Responds:** Suggestions accepted. The phrase has been corrected in the manuscript.

Title:

Changed to, "Study on the seasonal variation of Aeolus wind product performance over China using ERA5 and radiosonde data".

L10:

"the seasonality of Aeolus detection performance", changed to, "the seasonality of Aeolus wind product performance".

L47:

"the seasonal fluctuation of Aeolus detection performance", changed to, "the seasonal fluctuation of Aeolus wind product performance".

L51:

"the variation of Aeolus detection performance in different seasons", changed to, "the variation of Aeolus wind product performance in different seasons".

L190:

"the variation in Aeolus detection performance", changed to, "the variation in Aeolus wind product performance".

L218:

"the seasonal variation in Aeolus detection performance", changed to, "the seasonal variation in Aeolus wind product performance"

L296:

"the seasonal variation of Aeolus detection performance in China", changed to, "the seasonal variation of Aeolus wind product performance in China".

L331:

"Aeolus detection performance over China", changed to, "Aeolus wind product performance over

China".

2. You define the relative error as representative for the detection performance (product quality…). Other studies, like Martin et al. 2021, Baars et al., 2020, Witschas et al. 2020, Lux et al. 2020 all use the scaled Median Absolute Deviation (MAD) as a measure of the product quality, which should be closest to the random error of Aeolus winds, compared to the ECMWF model (as done in NWP monitoring). Your results would fit better in the selection of Aeolus validation studies, if you would also show the scaled MAD.

**Responds:** Suggestions accepted. Table 4 has been added below Figure 5 to show the scaled MAD. L171:

Added, "The MAD (Median Absolute Deviation) of different regions are showed in Table 4 to fit better with other studies. The MAD of the Aeolus wind data in a rectangle of ±1° lat/lon centered on the RS site are also calculated in Table 4, which have varying degrees of decline. The decrease of Shapingba group is not obvious because too few Aeolus data points meet the screening conditions (≤1° from RS site)."

L181:

Added,

| MAD(Aeolus&L) m/s | Chifeng | Baoshan | Shapingba | Qingyuan |
|---|---|---|---|---|
| Rectangle of ±2.5° | 4.48 | 4.42 | 4.47 | 4.32 |
| Rectangle of ±1° | 3.81 | 3.66 | 4.12 | 3.71 |

**Table 4. MAD(Aeolus & L) of different regions and different horizontal scales. The distance of 1° lat/lon is approximately 100km in the regions we studied.**

3. You applied a collocation distance of 2.5 deg lat/lon for valid overpasses. However, the cal/val implemtation plan (https://earth.esa.int/eogateway/documents/20142/1564626/Aeolus-Scientific-CAL-VAL-Implementation-Plan.pdf) suggests to use a horizontal collocation distance of 100 km around the ground site, which is much stricter than 2.5 deg. I understand that depending on the wind direction, the radiosonde might drift closer towards the actual Aeolus wind profile, but some further justifications of the 2.5 deg criteria would be desirable. Particularly for Mie-cloudy winds you can have significant deviations when using such a large distance.

**Responds:** Suggestions accepted. The 2.5 deg lat/lon was choosen to filter the Aeolus data for three reasons.

First and foremost, the Aeolus data points in 100 km around the RS site are too few for the study of statistical laws. Especially in the case of monthly research, there may be only a few data points per month, which result in unreliable statistical research results. It would be nearly impossible if we want to subdivide the vertical height of these data points. The number of Rayleigh-clear valid data points in different horizontal scales are shown in Table R1. The data points in Shapingba are reduced to only 0.1% of the original number.

| Number of Aeolus data points | Chifeng | Baoshan | Shapingba | Qingyuan |
|---|---|---|---|---|

| | | | | |
|---|---|---|---|---|
| Rectangle of ±2.5° | 19869 | 15283 | 22169 | 16685 |
| Rectangle of ±1° | 1753 | 3035 | 21 | 3367 |

**Table R1. The number of Rayleigh-clear valid data points in different horizontal scales.**

Secondly, the Chinese atmospheric model (GJB 5601-2006) used in our previous work provided monthly mean wind field with a grid resolution of 5° lat/lon ( http://www.cape.com.cn/ ).

Thirdly, the spatial difference of RS data with Aeolus data was also considered in original manuscript. ERA5 data were used as the reference data to solve this problem, as the error between ERA5 and RS was designed to approximately represent the temporal and spatial difference between RS and Aeolus.

The verification of the quality of Aeolus wind data points within 100 km from the RS site has been added, which is shown in the reply to the suggestion 2 (Table 4) and suggestion about *Lines 146-153* (Table 3). In addition, an explanation about the selection of collocation distance has been added in the manuscript at L56.

L56:

Added, "The collocation distance suggested by the cal/val implemtation plan (https://earth.esa.int /eogateway/documents/20142/1564626/Aeolus-Scientific-CAL-VAL-Implementation-Plan.pdf) is 100 km around ground site, but for a single site, there are too few data points that meet the suggestion per month."

**Detailed comments:**

*Line 15: Relative error 174% higher than…*

I would recommend to mention mean random wind errors of Aeolus against L-band radiosonde data. For external readers, a random error in m/s is more easily understandable than a %.

**Responds:** Suggestions accepted. The mean random wind errors of Aeolus against L-band radiosonde data of the four regions in July and December have been recalculated. It was added to the manuscript.

L15:

Added, "The mean random error increase 0.97m/s in July compared with December, which also supports this conclusion."

L192:

Added, "The mean random error is also calculated and has a 0.97m/s increase in July than December."

*Line 25: which uses a single-view detection method to scan the global three-dimensional wind field from space…*

ALADIN does not scan the 3D wind field, just the Horizontal line-of-sight (HLOS) component of the 3D wind field. Please rephrase.

**Responds:** Suggestions accepted. This expression is loosely and has been revised in our manuscript.

L25:

"which uses a single-view detection method to scan the global three-dimensional wind field from space", changed to, "which uses a single-view detection method to get the HLOS component of the

three-dimensional wind field from space".

*Line 32: Level 2B products provide HLOS wind data after actual atmospheric correction and deviation correction*
Better: fully processed HLOS wind profiles after correction of temperature and pressure effects.
**Responds:** Suggestions accepted. The expression has been changed in origin manuscript.
L32:
"Level 2B products provide HLOS wind data after actual atmospheric correction and deviation correction", changed to, "Level 2B products provide fully processed HLOS wind profiles after correction of temperature and pressure effects."

*Line 34: relevant researchers*
dedicated calibration and validation team have carried out ...
**Responds:** Suggestions accepted. The expression has been changed in origin manuscript.
L34:
"relevant researchers have carried out a series of verification and comparison studies on Aeolus' wind products", changed to, "dedicated calibration and validation team have carried out a series of verification and comparison studies on Aeolus' wind products."

*Line 48: After the implementation of the new M1 deviation correction scheme, the effect of system thermal performance changes on Aeolus' seasonal fluctuation is theoretically excluded.*
*...* is significantly reduced which led to systematic errors lower than 1 m/s (Rennie and Isaksen, 2020)
**Responds:** Suggestions accepted. The expression has been corrected in origin manuscript.
L48:
"After the implementation of the new M1 deviation correction scheme, the effect of system thermal performance changes on Aeolus' seasonal fluctuation is theoretically excluded", changed to, "After the implementation of the new M1 deviation correction scheme, the effect of system thermal performance changes on Aeolus' seasonal fluctuation is significantly reduced which led to systematic errors lower than 1 m/s (Rennie and Isaksen, 2020)."

*Line 55: Data and methods:*
I would recommend to rearrange the chapter just a little bit. Start with Aeolus L2B, Lband, ERA5 and put the description of the collocation criteria together with Fig. 1 under 2.4, when you explain the data matching.
**Responds:** Suggestions accepted. The chapter has been rearranged in the manuscript.
The content between lines 56 and 63 is migrated to line 96.

Line 76: What is the wind uncertainty of the L-band data. Please add a reference for the radiosonde wind bias here.
**Responds:** Suggestions accepted. Information about uncertainty of the L-band data have been added in L167, which has been introduced in *Author's Response to Referee #1, question 1.*

Line 85: You mention that ERA5 data is often used as reference in meteorological data analysis. However, we know that modelled wind predictions have high uncertainties too, particularly in cloudy situations. Please add some information of the expected model wind uncertainties in this chapter.

**Responds:** Suggestions accepted. As the quality of ERA5 wind data varies in different situations, we have added information about uncertainty of the ERA5 data in L167, which has been introduced in *Author's Response to Referee #1, question 1.*

*Line 99: Conversion from ERA5 to Aeolus HLOS:*

In Eq. 1, you use the wind direction and horizontal wind speed of the radiosonde data and the Aeolus azimuth angle to reproject the RS data to Aeolus data. I don't understand, why you do the same with the ERA5 data, where you have the full u and v wind vector information. Wouldn't be the correct conversion: HLOS = −u · sin(φ) − v · cos(φ) ?

**Responds:** Suggestions accepted. This is only for formal uniformity, and there is no difference between the results and what the referee said. In order not to mislead readers, it has been explained further in original manuscript.

L101:

Added, "The ERA5 wind data have full $\boldsymbol{u}$ and $\boldsymbol{v}$ wind victor information and can calculated the $V_{HLOS}$ through two components. However, the result is the same as that calculated by Eq. (1) after total wind vector composited."

*Line 115*: Please spell check the variable V_ture, it should be V_true I guess?

**Responds:** Suggestions accepted. The expression of Eq. (3) itself is ambiguous and has been clarified in *Author's Response to Referee #1, question 7.*

*Line 124*: You define the relative error as representative for the detection performance (product quality…). Other studies, like Martin et al. 2021, Baars et al., 2020, Witschas et al. 2020, Lux et al. 2020 all use the scaled Median Absolute Deviation (MAD) as a measure of the product quality, which should be closest to the random error of Aeolus winds, compared to the ECMWF model (as done in NWP monitoring). Your results would fit better in the selection of Aeolus validation studies, if you would also show a scaled MAD.

**Responds:** Suggestions accepted. This suggestion is same as the *suggestion 2* above and it has been responded here.

*Fig. 3*: It is recommended, also during several Aeolus validation workshops, to have HLOS Aeolus on the y-axis and all reference observations, models on the x-axis. Please swop Aeolus and L-band on the fig.3 (a,d). I would further advise to have Aeolus-L-band first, Aeolus-ERA5 in the middle, and ERA5-L band as last plot. You are describing (a,b…) but the letters are not shown in the plot.

**Responds:** Suggestions accepted. The Fig. 3 has been modified as suggestion. Meanwhile, the table of Fig. 3 have also been revised.

Fig. 3:

Changed to,

[Figure]

**Figure 3. Comparison results of Aeolus data, L-band RS data and ERA5 data in Mie-cloudy and Rayleigh-clear groups. (a,d) Aeolus vs L-band RS, (b,e) L-band RS vs ERA5 and (c,f) ERA5 vs Aeolus.**

Table 1:

Changed to,

|  | Aeolus vs L | | Aeolus vs ERA5 | | ERA5 vs L | |
|---|---|---|---|---|---|---|
|  | Mie-cloudy | Rayleigh-clear | Mie-cloudy | Rayleigh-clear | Mie-cloudy | Rayleigh-clear |
| R | 0.92 | 0.94 | 0.97 | 0.96 | 0.95 | 0.97 |
| N samples | 38275 | 73131 | 38275 | 73131 | 38275 | 73131 |
| Slop | 0.95 | 0.97 | 0.97 | 1.00 | 0.98 | 0.96 |
| Intercept(m/s) | -0.01 | -0.05 | 0.07 | 0.05 | -0.08 | -0.09 |
| MD(m/s) | -0.04 | -0.02 | 0.24 | 0.34 | 0.05 | 0.05 |
| SD(m/s) | 6.47 | 7.61 | 4.14 | 6.09 | 5.39 | 5.04 |

**Table 1. Comparison between Aeolus, L HLOS and ERA5 HLOS wind**

*Lines 146-153:* You are mentioning the space time matching problem. In the Cal/Val implementation plan, it is recommended to use a collocation criteria of +-100 km (which is much less than 2.5 deg) and +-60 min in time. One possibility could be to show the results for all measurements and one table for all within the 100 km distance. This may improve your results.

**Responds:** Suggestions accepted. One table has been added for all within the 1° lat/lon distance.

|  | Aeolus vs L | | Aeolus vs ERA5 | | ERA5 vs L | |
|---|---|---|---|---|---|---|
|  | Mie-cloudy | Rayleigh-clear | Mie-cloudy | Rayleigh-clear | Mie-cloudy | Rayleigh-clear |
| R | 0.96 | 0.95 | 0.97 | 0.96 | 0.98 | 0.99 |
| N samples | 4482 | 8067 | 4482 | 8067 | 4482 | 8067 |
| Slop | 0.96 | 0.99 | 0.97 | 1.00 | 0.99 | 0.98 |
| Intercept(m/s) | -0.22 | -0.02 | -0.01 | 0.19 | -0.22 | -0.19 |
| MD(m/s) | -1.0E-3 | -4.7E-5 | -6.1E-5 | 1.4E-3 | -9.8E-4 | -1.5E-3 |
| SD(m/s) | 4.73 | 6.18 | 4.19 | 6.10 | 3.09 | 2.97 |

**Table 2. Same as Table 1, but for rectangle of ±1° lat/lon. The distance of 1° lat/lon is approximately 100km in the regions we studied.**

L153:

Added, "The comparison results of data 1° from RS site are also shown in Table 2. Stricter collocation criteria led to better results."

*Fig. 5:* In order to compare it easier to the random error evolution, presented by Rennie and Isaksen, 2020, I would recommend to show the scaled MAD, additionally to the relative error. As mentioned before, other studies one the scaled MAD as the main estimate of the wind product random error.
**Responds:** Suggestions accepted. This suggestion is similar as the *suggestion 2* above and it has been responded here.

*Lines 194-196:* The fact that the relative errors are larger in summer 2020 compared to summer 2019 is mainly caused by the decrease in the output laser energy of ALADIN. This effect is less pronounced for Mie-cloudy winds, because they are not as strongly depended on the laser energy.
**Responds:** Suggestions accepted. The expression has been corrected in origin manuscript.
L195:
"This may be related to the differences in summer weather conditions in different years or the decrease in laser power of Aeolus. It is not yet clear.", changed to, "This is mainly caused by the decrease in the output laser energy of ALADIN. This effect is less pronounced for Mie-cloudy winds, because they are not as strongly depended on the laser energy."

Lines 205-210: On the other hand, the different situation of the Mie-cloudy group and the Rayleigh-clear group does not seem to support the explanation that the Aeolus system itself causes seasonal changes in detection performance…
What do you mean with that? Please add some further explanations on that. NWP monitoring results and other independent validation studies show that the Mie-cloudy bias is much less effected by thermal variations compared to the Rayleigh-clear bias. This can be explained by technical differences of the Mie spectrometer compared to the Rayleigh spectrometer.
**Responds:** Suggestions accepted. It has been clarified in origin manuscript.
L205:
"On the other hand, the different situation of the Mie-cloudy group and the Rayleigh-clear group does not seem to support the explanation that the Aeolus system itself causes seasonal changes in detection performance", changed to, "Seasonal variations in relative errors may be partly due to seasonal variations in atmospheric conditions".

*Line 216: The wind direction of the atmospheric wind field has an obvious seasonal trend.*
Can you provide a reference publication for this? I believe there are some manuscripts showing the seasonal trends of the wind direction in China.
**Responds:** Suggestions accepted. The references showing the seasonal trends of the wind direction in China was added.
L216:
Added references, "The wind direction of the atmospheric wind field has an obvious seasonal trend. In China, the northwest monsoon prevails in winter while the southeast in summer (Chang, 2004; HUANG et al., 2004; He et al., 2007)."

*Line 224-225: Based on the previous data matching work, we calculate the angle α between the real horizontal wind direction (provided by 225 ERA5 data) and the Aeolus HLOS direction (provided by L2B data) of each Aeolus valid data point…*

How do you calculate the wind direction from the HLOS data? Or do you mean that you use the azimuth angle, given in the L2B product?

**Responds:** Suggestions accepted. Yes, the Aeolus HLOS direction has the same meaning as the azimuth angle, and it has been clarified in manuscript.

L225:

Added, "the Aeolus HLOS direction (same as the azimuth angle provided by L2B data)"

*Line 240: …resulting in decrease in the energy of the laser beam and signal-to-noise ratio (SNR):*

Instead of decrease in energy, better say: "resulting in a decrease of the laser return signal and…"

**Responds:** Suggestions accepted. The expression has been corrected in origin manuscript.

L240:

"resulting in decrease in the energy of the laser beam and signal-to-noise ratio (SNR)", changed to, "resulting in a decrease of the laser return signal and signal-to-noise ratio (SNR)".

*Line 242: we takes the effective data*

we take the effective data…

**Responds:** Suggestions accepted. It has been corrected in origin manuscript. The manuscript has been thoroughly revised by Elsevier language editing services to avoid non-standard English expression.

L242:

"we takes the effective data", changed to, "we take the effective data of the Rayleigh channel."

Line 256: Mie-cloudy data does not distinguish between aerosol and clouds. You mention the impact of aerosols here, but some further comments about the potential impact of low-level aerosol layers on the wind product quality would be desirable. Could seasonal variations of aerosol layers also influence the results?

**Responds:** Suggestions accepted. Yes, this suggestion is insightful and the effects of aerosols on wind products will be studied in the following studies.

*Fig. 12:* Please ad (a) and (b) to the image.

**Responds:** Suggestions accepted. The picture has been corrected in origin manuscript.

Fig.12:

Changed to,

[Figure]

**Figure 12. Distribution of Mie-cloudy data points and LMNOPQR in altitude. (a) July 2020, (b) August 2020, and data are all from Baoshan.**

Line 319: Avoid to use double reasonable.

**Responds:** Suggestions accepted. It has been corrected in origin manuscript.

L319:

"This conjecture is also reasonable and can reasonably explain the seasonal variation in the relative error of the Mie-cloudy channel Aeolus.", changed to, "This conjecture is also reasonable and can explain the seasonal variation in the relative error of the Mie-cloudy channel Aeolus."

*311-320:* Have you thought about the effect of horizontal cloud variability on the Mie-cloudy wind errors? High clouds tend to be more homogeneous than low level clouds, which can vary strongly within one ~10 km Mie-cloudy wind result. This could potentially cause the higher relative errors in lower altitudes. Furthermore, the lidar signal is usually attenuated, if there are high clouds, thus I don't think that there will be Mie-cloudy valid winds from low altitude levels below high-altitude Mie-cloud valid winds.

**Responds:** Suggestions accepted. In the manuscript we mainly analyzed influence of high-altitude cloud and did not think about the effect of horizontal cloud variability on the Mie-cloudy wind errors. It will be involved in the following research.

*Line 323: Besides, as the first spaceborne wind lidar, the analysis of factors affecting the detection performance of Aeolus will help provide a reference for the follow-up development of spaceborne wind lidar.*

Could you be a bit more specific what of your findings will contribute to the follow-on development of future spaceborne wind lidars? Just add some more details here.

**Responds:** Suggestions accepted. More details has been added here.

L324:

"Besides, as the first spaceborne wind lidar, the analysis of factors affecting the detection performance of Aeolus will help provide a reference for the follow-up development of spaceborne

wind lidar", changed to, "Besides, this study is helpful to figure out the influence of clouds and wind direction on the detection performance of spaceborne wind lidar."

**References:**

Chang, C.-P.: East Asian Monsoon, World Scientific, 2004.

He, J., Ju, J., Wen, Z., Lü, J., and Jin, Q.: A review of recent advances in research on Asian monsoon in China, 24, 972–992, 2007.

HUANG, R., HUANG, G., and WEI, Z.: Climate variations of the summer monsoon over China, in: East Asian Monsoon, World Scientific, 213–268, 2004.

Rennie, M. and Isaksen, L.: The NWP impact of Aeolus Level-2B winds at ECMWF, ECMWF Technical Memo 864, Shinfield Park, Reading, UK, https://doi. org/10 …, 2020.

---

## Referee Report (RR1)

**Second response of referee #2 to the authors**

In this second review, the authors response is shown in black and referee's comments are shown in blue.

Thank you for your comprehensive revision. The manuscript has improved significantly in comparison to the previous version and is now acceptable for publication subject to some minor revisions, which are shown below.

L128: Calculation of relative errors: In this chapter, you are introducing all statistical measures to assess the wind product quality. I would like to see here also the equation of the scaled MAD, which you are showing later.

L204: Great that you now also calculate the MAD, you will see that your manuscript will be cited more often because the MAD (scaled MAD) is a much better variable for indicating the random wind error.
It would be even better, if you could remove the table here, and rather add the scaled MAD to Table 2 and Table 3, the overview of all errors for 2.5 deg and 1 deg. Please use the scaled MAD and not only the MAD. The scaled MAD is: MAD × 1.4826
Other Aeolus validation studies are all referring to the scaled MAD. Just multiply your result by 1.4826

L168: Thanks for adding an extra table for the 100 km distance. It clearly shows that the results are improved. I would recommend to also mention the numbers in the text. Something like: …. The correlation coefficient is increased and the standard deviation is decreased, when applying a stricter collocation criteria of 100 km.

L324:  "Besides, as the first spaceborne wind lidar, the analysis of factors affecting the detection performance of Aeolus will help provide a reference for the follow-up development of spaceborne
wind lidar", changed to, "Besides, this study is helpful to figure out the influence of clouds and wind direction on the detection performance of spaceborne wind lidar."
Now line L357: Thanks for updating the manuscript. However, I was suggesting to add some more information and not to remove the outlook to future developments of follow-on Doppler Wind Lidar missions. I like this outlook and would advise to take it back in the manuscript.

---

## Author Response (AR2)

**Author's Response to the Second Response of Referee #2**

In this response, the referee comments (in black) are listed together with our replies (in blue) and the changes to the original manuscript (in red).

Thanks a lot for your reviews on our manuscript. The comments are very helpful for revising and improving our paper. We have revised the manuscript according to the comments and the details are shown as follows.

L128: Calculation of relative errors: In this chapter, you are introducing all statistical measures to assess the wind product quality. I would like to see here also the equation of the scaled MAD, which you are showing later.

**Responds:** Suggestions accepted. The equation has been added in the manuscript.
L136:
added, "In addition, the scaled median absolute deviation (scaled MAD) is widely used in other Aeolus validation studies:

$$\text{scaled MAD} = 1.4826 \times \text{median}(|D - \text{median}(D)|), \tag{6}"$$

L204: Great that you now also calculate the MAD, you will see that your manuscript will be cited more often because the MAD (scaled MAD) is a much better variable for indicating the random wind error.

It would be even better, if you could remove the table here, and rather add the scaled MAD to Table 2 and Table 3, the overview of all errors for 2.5 deg and 1 deg. Please use the scaled MAD and not only the MAD. The scaled MAD is: MAD × 1.4826

Other Aeolus validation studies are all referring to the scaled MAD. Just multiply your result by 1.4826

**Responds:** Suggestions accepted. The scaled MAD has been added in the Table 2 and Table 3, while the Table 5 in L204 has been removed. Meanwhile, an error of MD calculation has been corrected.
L204:
Table 5 was removed.
L197:
removed,
"The MAD (Median Absolute Deviation) of different regions are showed in Table 5 to fit better with other studies. The MAD of the Aeolus wind data in a rectangle of ±1° lat/lon centered on the RS site are also calculated in Table 5, which have varying degrees of decline. The decrease of Shapingba group is not obvious because too few Aeolus data points meet the screening conditions (≤1° from RS site)."
L159:
Table 2, changed to,

|  | Aeolus vs L | | Aeolus vs ERA5 | | ERA5 vs L | |
|---|---|---|---|---|---|---|
|  | Mie-cloudy | Rayleigh-clear | Mie-cloudy | Rayleigh-clear | Mie-cloudy | Rayleigh-clear |
| R | 0.92 | 0.94 | 0.97 | 0.96 | 0.95 | 0.97 |
| N samples | 38275 | 73131 | 38275 | 73131 | 38275 | 73131 |
| Slop | 0.95 | 0.97 | 0.97 | 1.00 | 0.98 | 0.96 |
| Intercept(m/s) | -0.01 | -0.05 | 0.07 | 0.05 | -0.08 | -0.09 |

| | | | | | | |
|---|---|---|---|---|---|---|
| MD(m/s) | -0.03 | -0.03 | 0.05 | 0.05 | 0.21 | 0.24 |
| SD(m/s) | 6.47 | 7.61 | 4.14 | 6.09 | 5.39 | 5.04 |
| Scaled MAD(m/s) | 5.76 | 6.98 | 3.62 | 5.39 | 2.04 | 1.94 |

**Table 2. Comparison between Aeolus, L HLOS and ERA5 HLOS wind**

Table 3, changed to,

| | Aeolus vs L | | Aeolus vs ERA5 | | ERA5 vs L | |
|---|---|---|---|---|---|---|
| | Mie-cloudy | Rayleigh-clear | Mie-cloudy | Rayleigh-clear | Mie-cloudy | Rayleigh-clear |
| R | 0.96 | 0.95 | 0.97 | 0.96 | 0.98 | 0.99 |
| N samples | 4482 | 8067 | 4482 | 8067 | 4482 | 8067 |
| Slop | 0.96 | 0.99 | 0.97 | 1.00 | 0.99 | 0.98 |
| Intercept(m/s) | -0.22 | -0.02 | -0.01 | 0.19 | -0.22 | -0.19 |
| MD(m/s) | -0.24 | 2.5E-3 | -0.01 | 0.20 | -0.23 | -0.20 |
| SD(m/s) | 4.73 | 6.18 | 4.19 | 6.10 | 3.09 | 2.97 |
| Scaled MAD(m/s) | 4.11 | 5.42 | 3.55 | 5.26 | 2.54 | 2.30 |

**Table 3. Same as Table 2, but for rectangle of ±1° lat/lon. The distance of 1° lat/lon is approximately 100km in the regions we studied.**

L165:

"However, there was no significant difference between the comparison results of the L-band RS and ERA5 and the comparison results of the Aeolus and ERA5. Therefore, the Aeolus and ERA5 data were in good agreement, and the error caused by the space-time matching problem of the L-band RS data might be larger than expected."

changed to, "In addition, the scaled MAD value of group ERA5 vs L was significantly lower than that of other groups. This means that there is a great agreement between ERA5 wind and L-band RS wind, despite their temporal and spatial matching problems."

L168: Thanks for adding an extra table for the 100 km distance. It clearly shows that the results are improved. I would recommend to also mention the numbers in the text.

Something like: …. The correlation coefficient is increased and the standard deviation is decreased, when applying a stricter collocation criteria of 100 km.

**Responds:** Suggestions accepted. More details about the numbers has been added in manuscript.

L168:

added, "The correlation coefficient is increased and the scaled MAD is decreased, when applying a stricter collocation criteria of 100 km."

Now line L357: Thanks for updating the manuscript. However, I was suggesting to add some more information and not to remove the outlook to future developments of follow-on Doppler Wind Lidar missions. I like this outlook and would advise to take it back in the manuscript.

**Responds:** Suggestions accepted. We have taken the outlook back in the manuscript.

L357:

"Besides, this study is helpful to figure out the influence of clouds and wind direction on the

detection performance of spaceborne wind lidar"

changed to, "Besides, this study is helpful to figure out the influence of clouds and wind direction on the wind product performance of Aeolus, which will provide a reference for the follow-up development of spaceborne wind lidar."